# Genetic architecture of natural variation in cuticular hydrocarbon composition in *Drosophila melanogaster*

Lauren M Dembeck[1,2,3†], Katalin Böröczky[2,3,4‡], Wen Huang[1,2,3], Coby Schal[2,3,4], Robert R H Anholt[1,2,3], Trudy F C Mackay[1,2,3*]

[1]Department of Biological Sciences, North Carolina State University, Raleigh, United States; [2]Genetics Program, North Carolina State University, Raleigh, United States; [3]W. M. Keck Center for Behavioral Biology, North Carolina State University, Raleigh, United States; [4]Department of Entomology, North Carolina State University, Raleigh, United States

**Abstract** Insect cuticular hydrocarbons (CHCs) prevent desiccation and serve as chemical signals that mediate social interactions. *Drosophila melanogaster* CHCs have been studied extensively, but the genetic basis for individual variation in CHC composition is largely unknown. We quantified variation in CHC profiles in the *D. melanogaster* Genetic Reference Panel (DGRP) and identified novel CHCs. We used principal component (PC) analysis to extract PCs that explain the majority of CHC variation and identified polymorphisms in or near 305 and 173 genes in females and males, respectively, associated with variation in these PCs. In addition, 17 DGRP lines contain the functional *Desat2* allele characteristic of African and Caribbean *D. melanogaster* females (more 5,9-C27:2 and less 7,11-C27:2, female sex pheromone isomers). Disruption of expression of 24 candidate genes affected CHC composition in at least one sex. These genes are associated with fatty acid metabolism and represent mechanistic targets for individual variation in CHC composition.

*For correspondence: trudy_mackay@ncsu.edu

Present address: †Ecology and Evolution Unit, Okinawa Institute of Science and Technology, Onna-son, Japan; ‡Department of Neurobiology and Behavior and Department of Ecology and Evolutionary Biology, Cornell University, Ithaca, United States

Competing interests: The authors declare that no competing interests exist.

## Introduction

Insects comprise the most species-rich class in the animal kingdom. They evolved about 480 million years ago and their fecundity and rapid evolutionary adaptations have made them successful in populating almost every ecological niche on our planet. The evolution of mechanisms for desiccation resistance was critical for insects to colonize dry land. To prevent desiccation insects evolved the ability to produce and accumulate species-specific blends of fatty acid-derived apolar lipids on the epicuticle; the most prominent of these are cuticular hydrocarbons (CHCs) (*Jallon et al., 1997*). CHCs are produced continuously by specialized cells called oenocytes, and are transported through the hemolymph and then to the cuticular surface through specialized pore canals (*Romer, 1991*; *Schal et al., 1998*; *Blomquist and Bagnères, 2010*). The primary role of CHCs is desiccation resistance (*Gibbs, 1998*; *2002*), but they have been co-opted to serve as chemical signals and cues mediating intra- and inter-specific social interactions (*Venard and Jallon, 1980*; *Jallon, 1984*; *Ferveur, 2005*). These interactions include species and nest-mate recognition, assessment of reproductive status, and mate choice, and CHCs play a prominent role in camouflage and mimicry that mediate inter-specific parasitic relationships (*Stanley-Samuelson and Nelson, 1993*; *Blomquist and Bagnères, 2010*).

Adaptive evolution acts through selection on phenotypic variation within a population. Thus, understanding the genetic basis for individual variation in CHC composition will shed light on

**eLife digest** The outermost layer of an insect's body is called the epicuticle and is made of a blend of fat molecules. "Cuticular hydrocarbons" (or CHCs) are the most common fat molecules in the epicuticle, and play an important role in protecting the insect's body from harsh, dry habitats. CHCs also have other roles in insect behavior. For example, these molecules act as chemical cues when insects search for mates (i.e. pheromones), and they can even contribute to camouflage.

Insects are amongst the most diverse groups of animals on Earth, and different species have different blends of CHC molecules in their epicuticles. Fruit flies are a useful model to understand the genetics of CHC production, including CHCs that act as sex pheromones. Previous research has analyzed the CHCs made by both sexes in several fruit fly strains. However this work was unable to uncover which genes influence how much of a given CHC an individual fly will make.

Dembeck et al. have now looked into CHC production in a collection of 205 different fly strains, all of which have already had their total genetic material sequenced and studied. Comparing these known sequences and looking for associations between genetic differences and particular CHCs uncovered 24 genes that may be involved in CHC manufacture. Only six of the genes had been identified previously. Dembeck et al. found that interfering with the activity of any of the 24 genes had a knock-on effect on many other CHCs present in the flies' epicuticle.

These 24 genes could to be pieced together in a network that is needed to make and recycle CHCs. The complexity and flexibility of this network can explain in part how insects have been able to build epicuticles for almost every environment. These data set the stage for future work directed towards understanding the evolutionary significance of variation in CHC composition in many fruit fly populations.

evolutionary mechanisms of assortative mating (*Noor et al., 1996*; *Ishii et al., 2001*; *Rundle et al., 2005*; *Gleason et al., 2005*) and evolution of social organization (*Howard and Blomquist, 2005*; *Richard and Hunt, 2013*).

The extensive genetic resources available for *Drosophila melanogaster* make it a valuable model for studying the genetic basis of CHC production and natural variation in CHC composition. Mature *D. melanogaster* have sexually dimorphic CHCs ranging from chain lengths of 21 to 31 carbons (C21–C31) (*Antony and Jallon, 1982*; *Jallon and David, 1987*). Males produce predominantly shorter-chain CHCs (< C26) and they use two of the CHCs, 7-C23:1 and 7-C25:1, as sex pheromones. Females produce predominantly longer-chain dienes, among which 7,11-C27:2 and 7,11-C29:2 act as the primary female sex pheromone components (*Antony and Jallon, 1982*; *Cobb and Jallon, 1990*; *Arienti et al., 2010*).

CHCs are produced from the fatty acid biosynthetic pathway, which begins with an acetyl-CoA. Acetyl-CoA carboxylase (ACC) then catalyzes the synthesis of malonyl-CoA, and the multifunctional protein fatty acid synthase (FASN) successively incorporates malonyl-CoA units onto the acetyl-CoA, elongating the chain by two carbons each time and forming long chain fatty acids (LCFA). RNAi-knockdown of *ACC* in the oenocytes completely eliminates CHCs in both male and female *D. melanogaster* (*Wicker-Thomas et al., 2015*). Products of insect FASN, including *Drosophila* FASN, can be 14, 16 or 18 carbon fatty acids. A thioesterase that is part of the multienzyme FASN removes the elongated chain as a free fatty acid, and fatty acid elongation and desaturation use the CoA derivative and take place in the microsomal fraction (endoplasmic reticulum). Members of a family of tissue-specific elongases (ELOVL) catalyze the incorporation of malonyl-CoA units to form very long chain fatty acids (VLCFA). After condensation of the malonyl-CoA with the fatty acyl-CoA the next three steps in each elongation cycle include reduction of a carbonyl to an alcohol (by a 3-keto-acyl-CoA-reductase; KAR), dehydration (by a 3-hydroxy-acyl-CoA-dehydratase; HADC), and reduction of the carbon-carbon double bond by a trans-enoyl-CoA-reductase (TER). The VLCA as the CoA derivative is readuced to an aldehyde by a fatty acyl-CoA reductase (FAR). The one-carbon chain-shortening conversion of aldehydes to hydrocarbons is catalyzed by a cytochrome P450 enzyme, Cyp4G1 (*Qiu et al., 2012*). Desaturation reactions to introduce double bonds, leading to unsaturated fatty acids, appear to occur on the 16 or 18 carbon fatty acyl-CoAs.

A comprehensive analysis of CHCs in both sexes segregating in a panel of recombinant inbred lines derived from a natural population identified 25 quantitative trait loci (QTL) in females and 15 in males contributing to variation in CHCs (*Foley et al., 2007*), but this study did not have the power to resolve QTLs to individual genes. QTL mapping analyses have also identified genomic regions called *small monoene quantities (smoq)* and *seven pentacosene (sept)*, respectively, associated with variation in the proportions of 7-C23:1 and 7-C25:1 in males (*Ferveur and Jallon, 1996*). In a mutagenesis study, another genomic region, *nerd,* drastically reduced 7-C23:1 production in males and altered courtship behavior (*Ferveur and Jallon, 1993*). However, none of these male loci have been resolved to specific genes. Several additional genes affecting CHC biosynthesis have been described in *D. melanogaster*: an acetyl-CoA carboxylase (*ACC*) (*Wicker-Thomas et al., 2015*); two fatty acid synthases, *FASN2 (CG3524)* which is involved in the synthesis of precursors of 2-methylalkanes (*Chung et al. 2014*), and *FASN1 (CG3523)* which is expressed in the fat body but nonetheless contributes to the CHC pool (*Wicker-Thomas et al., 2015*); an elongase (*EloF,* Wicker-Thomas et al., 1997; *Chertemps et al., 2007*); *KAR (CG1444), TER (CG10849)* and *HADC (CG6746)* whose knockdown eliminates or reduces CHC synthesis (*Wicker-Thomas et al., 2015*); a cytochrome P450 (*Cyp4G1* [*CG3972*], *Qiu et al., 2012*); and three desaturases, *Desat1, Desat2* and *DesatF* (*Labeur et al., 2002; Marcillac et al., 2005; Chertemps et al., 2006; Fang et al., 2009*).

Here, we used the sequenced, inbred lines of the *D. melanogaster* Genetic Reference Panel (DGRP) (*Mackay et al., 2012; Huang et al., 2014*) to perform genome wide association (GWA) analyses for nearly all detectable CHCs in both sexes in a scenario where all common genetic variants are genotyped and local linkage disequilibrium (LD) is sufficiently low to identify candidate genes and causal polymorphisms. We found considerable heritable genetic variation in a majority of male and female CHCs, distilled the axes of genetic variation into several principal components (PCs), and performed GWA analyses on each PC. We identified 24 candidate genes plausibly associated with CHC biosynthesis and for all of them disruption of their expression altered CHC profiles in males, females, or both sexes. Surprisingly, we also found that the DGRP lines are segregating for the ancestral and deletion alleles in *Desat2,* previously associated with CHC profiles thought to be unique for African flies. Finally, our results provide a new perspective and highlight the complexity of the biosynthetic and catabolic pathways that regulate the dynamics of CHC composition and provide the stage for adaptive evolution. In agreement with other recent studies of complex traits, our results demonstrate that the genetic architecture underlying potentially adaptive traits can consist of many, even hundreds, of polymorphic loci with small effects affecting different aspects of the phenotype.

## Results

### CHCs in the DGRP

In this article, linear alkanes are referred to by the abbreviation *n*-Cx, where ''x'' is the total carbon number. For example, tricosane is designated *n*-C23. Methyl-branched alkanes are referred to with a y-Me-Cx prefix, where 'y' is the carbon onto which the methyl group is bound. For example, 9-Me-C23 is a 23 carbon chain with a methyl on the $9^{th}$ carbon. For monoenes (z-Cx:1) and dienes (z,z-Cx:2) the number of double bonds is indicated after the colon and the double bond position/s are 'z' or 'z,z"). For example, 7-C23:1 has one double bond between the $7^{th}$ and $8^{th}$ carbons. We identified and quantified 71 female CHCs and 42 male CHCs in 169 and 157 DGRP lines, respectively (*Table 1*, *Figure 1*, *Supplementary file 1*). Sixteen of these CHCs have not been described previously in *D. melanogaster*. Eight of the new compounds were methyl-branched CHCs, seven were dienes, and one was a monoene. Nine of these compounds were detected only in females.

We assessed the extent to which the CHCs were genetically variable in the DGRP. All but three female CHCs and one male CHC (female peaks 26, 39, and 52 and male peak 59) showed significant among-line variation in a univariate ANOVA (*Supplementary file 2*). Broad sense heritabilities in females ranged from 0.98 for 7,11-C25:2 (peak 24) to 0.22 for 6-C25:1 (peak 30). Broad sense heritabilities for males ranged from 0.97 for 7-C25:1 (peak 29) to 0 for 9-C29:1 (peak 58; *Supplementary file 2*).

**Table 1.** Cuticular lipids identified by GC-MS in DGRP males and females. NI = not identified; nd = not detected; bold typeface = not previously identified in *D. melanogaster*.

| # | Cuticular component | Retention index ♀ | Retention index ♂ | # | Cuticular component | Retention index ♀ | Retention index ♂ |
|---|---|---|---|---|---|---|---|
| 1 | *n*-C21 | 2100 | 2100 | 33 | NI | 2516 | nd |
| 2 | *x*-C22:1 (quantified only in ♂) | 2179 | 2179 | 34 | NI | 2521 | nd |
| 3 | *x*-C22:1 | nd | 2184 | **35** | **13-Me-C25** **11-Me-C25** | 2533 | nd |
| c | cis-vaccenyl acetate | nd | 2189 | **36** | **5-Me-C25** | 2550 | nd |
| 4 | *n*-C22 | 2200 | 2200 | **37** | **8,12-C26:2** | 2555 | nd |
| 5 | 7,11-C23:2 | 2259 | nd | 38 | 7,11-C26:2 | 2560 | nd |
| 6 | 2-Me-C22 | 2263 | 2263 | 39 | 2-Me-C25 | 2562 | 2562 |
| 7 | NI | nd | 2267 | 40 | 6,10-C26:2 | 2566 | nd |
| 8 | 9-C23:1 | 2273 | 2273 | **41** | 9-C26:1 (only in ♀) **3-Me-C25** | 2572 | 2572 |
| 9 | 7-C23:1 | 2280 | 2283 | 42 | 7-C26:1 | 2577 | 2577 |
| 10 | 6-C23:1 | 2285 | 2286 | 43 | 6-C26:1 + impurity (i) | 2581 | 2581 |
| 11 | 5-C23:1 | 2291 | 2291 | 44 | *n*-C26 | 2600 | 2600 |
| 12 | *x*-C23:1 | 2294 | nd | 45 | 9,13-C27:2 | 2652 | nd |
| 13 | *n*-C23 | 2300 | 2300 | 46 | 7,11-C27:2 (only in ♀) 2-Me-C26 | 2664 | 2663 |
| 14 | **11-Me-C23** **9-Me-C23** | 2336 | 2336 | 47 | 5,9-C27:2 (only in ♀) 9-C27:1 | 2675 | 2675 |
| 15 | 7,11-C24:2 | 2355 | nd | 48 | 7-C27:1 | 2682 | 2682 |
| 16 | ***x,y*-C24:2** | 2363 | 2363 | 49 | 5-C27:1 | 2693 | nd |
| 17 | 3-Me-C23 9-C24:1 (only in ♀) | 2373 | 2373 | 50 | *n*-C27 | 2700 | 2700 |
| 18 | 7-C24:1 (quantified only in ♂) | 2377 | 2377 | **51** | **8,12-C28:2** | 2756 | nd |
| 19 | 6-C24:1 | 2380 | 2380 | 52 | 2-Me-C27 7,11-C28:2 (only in ♀) | 2761 | 2761 |
| 20 | NI (Variable in DGRP lines but not detected in GC-MS samples) | nd | nd | **53** | **6,10-C28:2** **3-Me-C27** 9-C28:1 | 2768 | nd |
| 21 | 5-C24:1 | 2386 | 2386 | 54 | NI | 2772 | 2772 |
| 22 | *n*-C24 | 2400 | 2400 | 55 | *n*-C28 | 2800 | 2800 |
| 23 | 9,13-C25:2 | 2451 | nd | 56 | 9,13-C29:2 | 2852 | nd |
| 24 | 7,11-C25:2 | 2460 | nd | 57 | 2-Me-C28 7,11-C29:2 (only in ♀) | 2864 | 2862 |
| 25 | 2-Me-C24 | 2463 | 2463 | 58 | 9-C29:1 5,9-C29:1 (only in ♀) | 2875 | 2875 |
| 26 | **6,10-C25:2** | 2468 | 2468 | 59 | 7-C29:1 | 2882 | 2882 |
| 27 | 5,9-C25:2 (only in ♀) 9-C25:1 | 2474 | 2474 | 60 | *n*-C29 | 2900 | 2900 |
| 28 | 8-C25:1 | 2478 | nd | **61** | **8,12-C30:2** (only in ♀) **2-Me-C29** | 2961 | 2961 |
| 29 | 7-C25:1 | 2482 | 2483 | 62 | 2-Me-C30 | 3060 | 3060 |
| 30 | **6-C25:1** | 2485 | nd | 63 | 7,11-C31:2 | 3065 | nd |
| 31 | 5-C25:1 | 2492 | 2492 | 64 | *n*-C31 | 3100 | 3100 |
| 32 | *n*-C25 | 2500 | 2500 | IS | *n*-C32 | 3200 | 3200 |

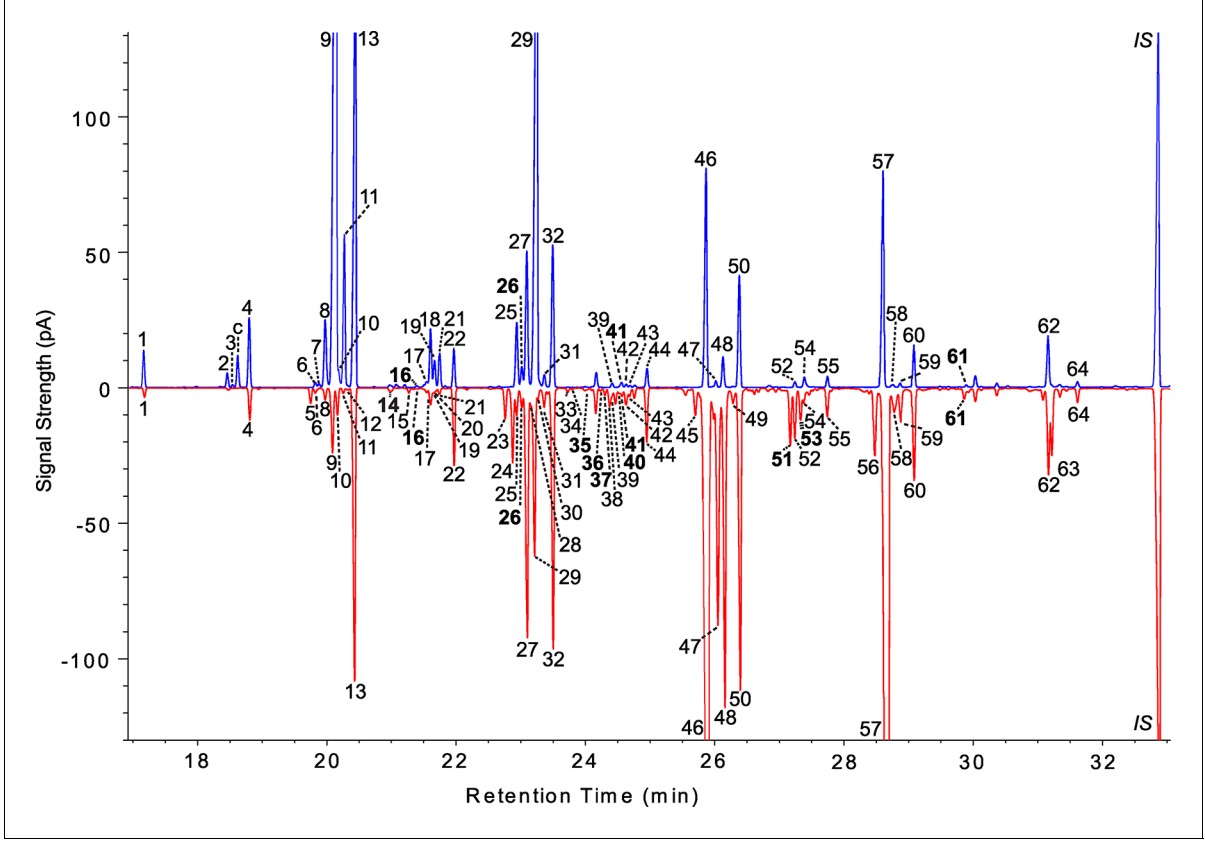

**Figure 1.** Representative male and female chromatograms from the DGRP. Male cuticular lipids of DGRP_38 are shown on the top (blue) and female CHCs of DGRP_786 are mirrored below (red). All peaks for both sexes were assigned a unique number based on its corresponding compound determined by GC-MS; thus compounds shared between the sexes carry the same number. See *Table 1* for the list of compound names. Compounds not previously described in *D. melanogaster* are shown in bold typeface. Some CHC isomers were not resolved by conventional GC, so a few chromatogram peaks contain more than one CHC. pA = picoAmperes, c = *cis*-vaccenyl acetate, * = contaminants from CHC extraction, IS = internal standard (*n*-C32).

## African female CHC phenotypes

The *Desat* locus was the first desaturase gene sequence described in insects (*Wicker-Thomas et al., 1997*) and has been implicated in the biosynthesis of pheromones. It consists of two desaturase genes, *Desat1* and *Desat2*. *Desat1* is expressed in both sexes and encodes a Δ-9 desaturase that catalyzes the synthesis of palmitoleic acid, an ω-7 fatty acid and precursor to 7-monoene and the first double bond of 7,11-dienes (*Labeur et al., 2002*; *Marcillac et al., 2005*). *Desat2* has a female-specific effect on CHC production and has been associated with adaptive divergence of African and Cosmopolitan races of *D. melanogaster* (*Greenberg et al., 2003*, but see *Coyne and Elwyn, 2006*). *Desat2* encodes a functional desaturase in African females, but is inactive in Cosmopolitan females due to a 16-bp deletion in the promoter region (*Coyne et al., 1999*; *Dallerac et al., 2000*, *Takahashi et al., 2001*). Females with an intact *Desat2* gene produce altered CHC profiles which are high in the pheromonal CHC isomers, 5,9-C27:2 and 5,9-C29:2, and low in the 7,11-isomers. African *D. melanogaster* females also exhibit a strong behavioral bias against non-African males (*Fang et al., 2002*; *Takahashi and Ting, 2004*; *Grillet et al., 2012*).

Surprisingly, females of 15 DGRP lines expressed the African phenotype; *i.e.*, they had high levels of 5,9-C27:2 and low levels of the primary female sex pheromone, 7,11-C27:2 (*Figure 2A*). Since *Desat2* (cytological position 87B10) is located within the common African inversion *In3R(K)* (computed breakpoints: 86F1-86F11;96F11-96F14 (*St. Pierre et al., 2014*) we expected the *Desat2* allele status (ancestral African or a Cosmopolitan 16-bp deletion in the promoter), CHC phenotype and inversion status to perfectly co-segregate in the DGRP lines. However, this was not the case. A total

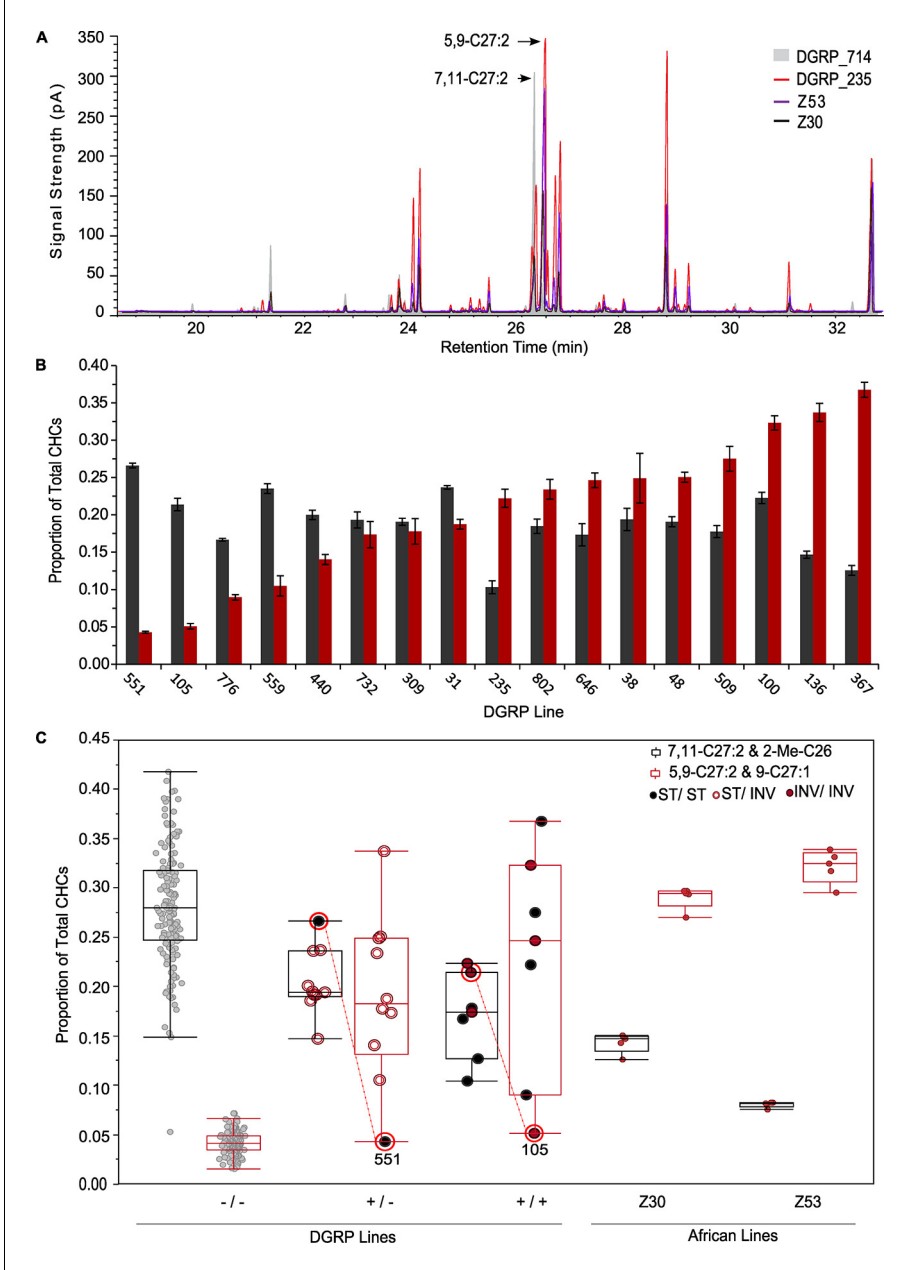

**Figure 2.** DGRP lines segregate for the female African CHC phenotype, *Desat2* allele, and *In3R(K)* inversion status. (A) Overlaid chromatograms of African *D. melanogaster* CHCs (Z30 and Z53), a DGRP line with an African-like CHC phenotype (DGRP_235), and a Cosmopolitan DGRP line (DGRP_714). (B) DGRP lines with at least one ancestral *Desat2* allele exhibit natural variation in the percentage of each CHC peak for the isomeric sex pheromones 7,11-C27:2 (2-Me-C26 co-elutes with 7,11-C27:2) (gray) and 5,9-C27:2 (9-C27:1 co-elutes with 5,9-C27:2) (red). (C) Box-plots of the proportion of each sex pheromone peak for DGRP and African lines according to *Desat2* allele and *In3R(K)* genotypes. DGRP_105 and DGRP_551, which have more Cosmopolitan-like phenotypes despite having the functional ancestral *Desat2* allele, are indicated.

of 17 DGRP lines with female CHC phenotypes contained the ancestral, functional *Desat2* allele (*Table 2*, *Figure 2B*). There was significant variation in the 7,11- and 5,9-C27:2 peaks among these lines (5,9-C27:2 & 9-C27:1 F = 35.17, p<0.0001; 7,11-C27:2 & 2-Me-C26 F = 16.09, p<0.0001; *Supplementary file 3*). We obtained female CHC data for two African lines, Z53 and Z30, for comparison. Females had significantly less 7,11-C27:2 according to deletion status and correspondingly more 5,9-C27:2 (*Figure 2C*). Females from DGRP lines that were either homozygous or

**Table 2.** Phenotypes and *In(3R)K* genotypes for females from DGRP lines with functional *Desat2* alleles. Red text indicates "mismatched" *Desat2* genotype ('+' = ancestral; '-' = 16-bp deletion) and inversion status ('INV' = *In(3R)K*; 'ST' = Standard karyotype). Blue background indicates "mismatched" *Desat2* genotype and phenotype.

| DGRP line | *Desat 2* genotype | % 7,11-C27:2 | % 5,9-C27:2 | Ratio | *In(3R)K* status |
|---|---|---|---|---|---|
| DGRP_31 | + / - | 23.7 | 18.7 | 1.27 | INV / ST |
| DGRP_38 | + / - | 19.4 | 24.9 | 0.78 | INV / ST |
| DGRP_48 | + / - | 19.1 | 25.0 | 0.76 | INV / ST |
| DGRP_100 | + / + | 22.3 | 32.3 | 0.69 | INV / INV |
| DGRP_136 | + / - | 14.7 | 33.7 | 0.44 | INV / ST |
| DGRP_309 | + / - | 19.1 | 17.8 | 1.07 | INV / ST |
| DGRP_440 | + / - | 20.0 | 14.0 | 1.43 | INV / ST |
| DGRP_559 | + / - | 23.5 | 10.5 | 2.24 | INV / ST |
| DGRP_646 | + / + | 17.3 | 24.6 | 0.70 | INV / INV |
| DGRP_802 | + / - | 18.5 | 23.4 | 0.79 | INV / ST |
| DGRP_732 | + / - | 19.3 | 17.3 | 1.12 | INV / ST |
| DGRP_367 | + / + | 12.6 | 36.8 | 0.34 | ST/ ST |
| DGRP_776 | + / + | 16.7 | 8.98 | 1.86 | ST / ST |
| DGRP_509 | + / + | 17.8 | 27.5 | 0.65 | ST / ST |
| DGRP_235 | + / + | 10.3 | 22.2 | 0.46 | ST / ST |
| DGRP_551 | + / - | 26.6 | 4.29 | 6.20 | ST / ST |
| DGRP_105 | + / + | 21.4 | 5.10 | 4.20 | INV / INV |

heterozygous for the ancestral *Desat2* sequence had intermediate amounts of the 7,11- and 5,9-C27:2 compared to the DGRP lines homozygous for the deletion or the African lines, respectively.

The co-segregation of the functional allele and *In3R(K)* was not perfect: five of the lines with the ancestral *Desat2* sequence were homozygous for the standard karyotype (*Figure 2C*, *Table 2*). Furthermore, two lines with the ancestral and presumably functional *Desat2*, DGRP_105 and DGRP_551, did not exhibit the African CHC phenotype. In total, six of the 17 lines with the ancestral sequence had mismatched inversion and CHC status in females. It is possible that the lines with homozygous standard karyotypes are actually segregating for the inversion at low frequency and thus only the homozygous standard flies were sampled for karyotyping. Alternatively, the *Desat2* deletion may have occurred prior to the inversion event.

We next checked *Desat2* for potentially damaging genetic variants (*Mackay et al., 2012*; *Huang et al., 2014*). We identified five *Desat2* alleles unique to DGRP_105. Two variants are synonymous coding (*3R_8262545_SNP* and *3R_8263020_SNP*), one is a deletion causing a frameshift (*3R_8263023_DEL*), and two are nonsynonymous coding (*3R_8263031_MNP* and *3R_8263048_SNP*). The frameshift and nonsynonymous variants are potentially damaging and could explain why this line produced the Cosmopolitan phenotype despite having the functional *Desat2* allele. We did not find such evidence for DGRP_551, suggesting this line may contain unknown genetic variants that inhibit the production of 5,9-C27:2.

## CHC correlations and principal components analyses

Most of the CHCs belong to homologous series in which the chain length increases by two carbons; thus, these compounds may be genetically correlated due to a shared biosynthetic pathway and the data may be confounded by multi-colinearity (*Martin and Drijfhout, 2009*). We visualized the correlations between CHCs using modulated modularity clustering (MMC) (*Stone and Ayroles, 2009*). The MMC algorithm clusters highly correlated variables based on the Spearman's rank correlation coefficients ($\rho$). As expected, some CHCs were highly correlated within each sex (*Figures 3* and *4*; *Supplementary file 4*).

In the first two female modules, seven dienes had strong positive correlations with each other. There was also one peak (47 – 5,9-C27:2 and 9-C27:1) that strongly negatively correlated with those dienes and the shorter-chain (≤ C25) alkanes of module 3 (*Figure 3*, *Supplementary file 4*). Similarly, the module 3 shorter-chain alkanes had strong positive correlations with each other, some dienes of modules 1 and 2, and monoenes and dienes in module 5. These four modules (1, 2, 3, and 5) all had weak to moderate negative correlations with module 7, which consisted of strongly inter-correlated longer-chain (≥ C25) alkanes and dienes.

We found similar trends in the male CHCs (*Figure 4*; *Supplementary file 4*). Module 1 consisted of longer-chain alkanes that negatively correlated with the shorter-chain alkanes of module 2. This

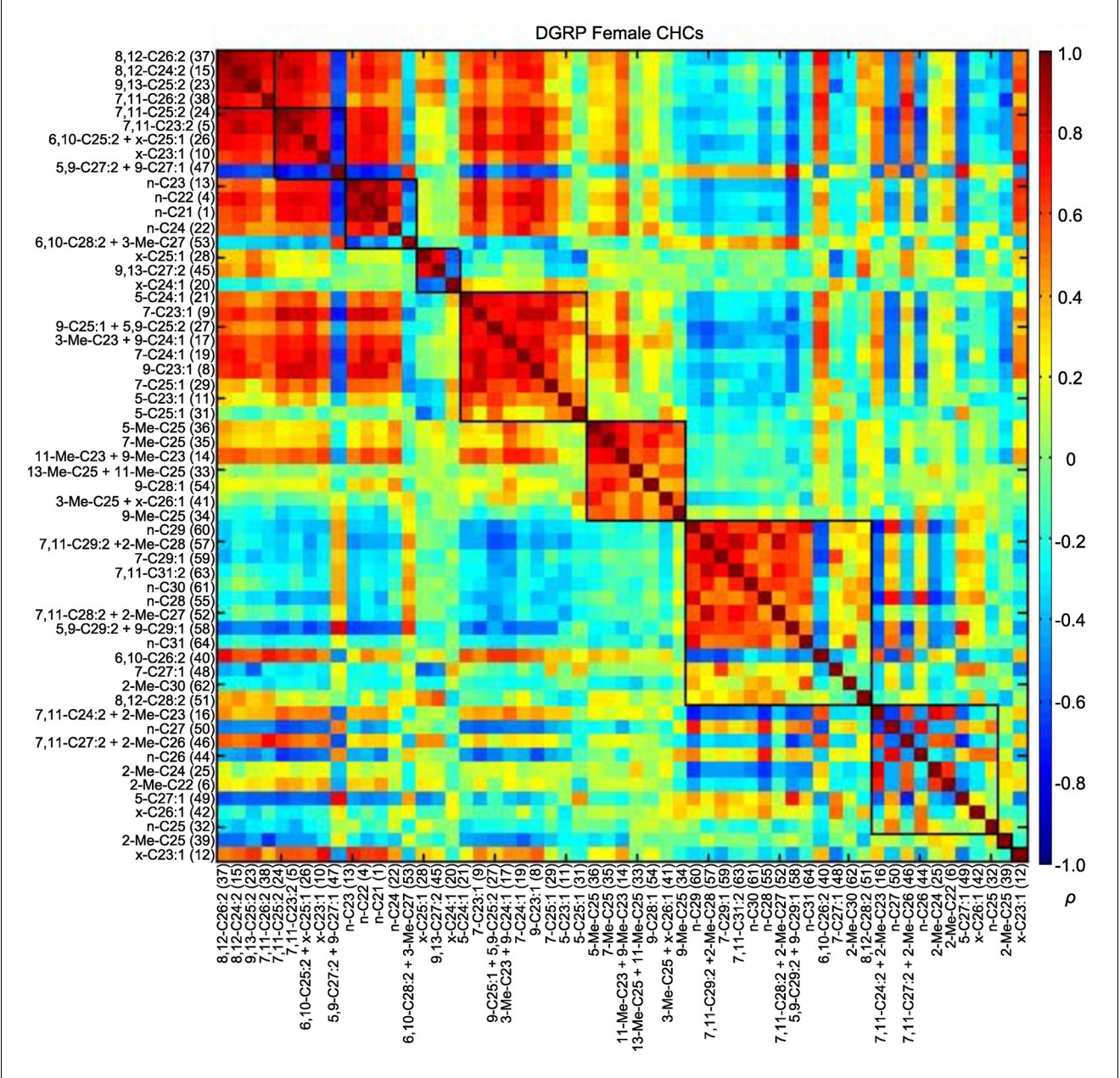

**Figure 3.** MMC modules of DGRP female CHCs based on Spearman's rank correlation coefficients ($\rho$). Correlations are color-coded from +1 (dark red) to -1 (dark blue). Correlated CHCs are clustered into groups (modules). Modules (outlined in black) are arranged along the diagonal according to the average strengths of the correlations within each cluster; the most strongly correlated modules are on the top left and the weakly correlated modules are on the bottom right.

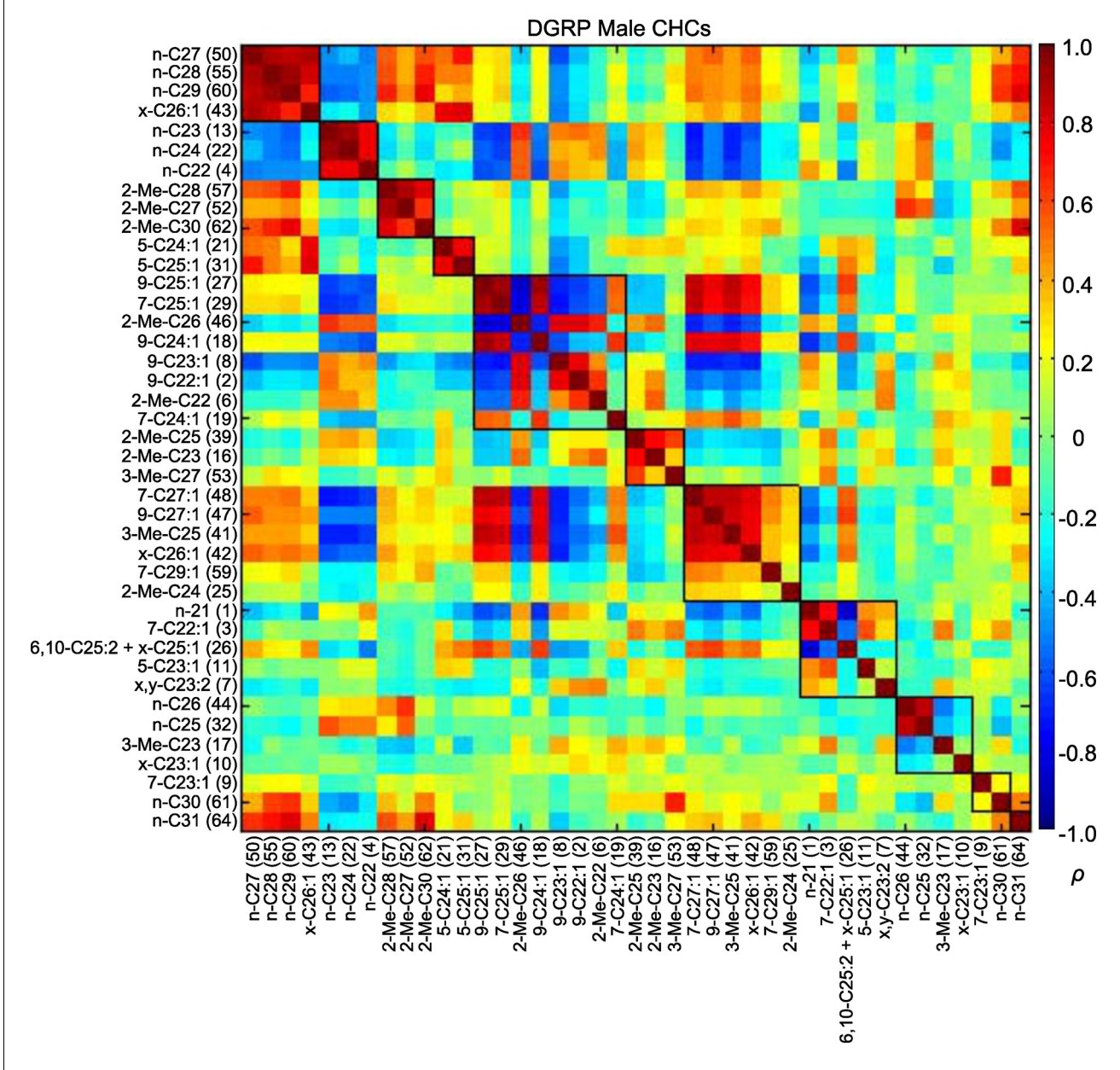

**Figure 4.** MMC modules of DGRP male CHCs based on Spearman's rank correlation coefficients ($\rho$). Correlations are color-coded +1 (dark red) to -1 (dark blue). Correlated CHCs are clustered into groups (modules). Modules (outlined in black) are arranged along the diagonal according to the average strengths of the correlations within the groups; the most strongly and weakly correlated are on the top left and bottom right, respectively.

could also be seen between module 2 and the longer-chain monoenes in module 7. These negative correlations were a consistent trend between other long- and short-chain CHCs and exemplify the tradeoff between short- and long-chain compounds, as the latter are produced through the elongation of fatty acids that serve as precursors for the former.

The correlations among CHCs are plausible given the biology of CHC production. We computed PCs of genetically variable CHCs within each sex to reduce the dimensionality of the data to orthogonal PCs. The first seven and five PCs accounted for 98.00% and 98.12% of the total variation among the DGRP lines for female and male CHCs, respectively (*Table 3*, *Figure 5*, *Supplementary file 5*). We hypothesized that the GWA results would provide insights into the genetic architecture underlying the MMC trends.

**Table 3.** Percent of CHC variation in the DGRP explained by PCs.

| Sex | Number | Eigenvalue | Percent | Cumulative percent |
|---|---|---|---|---|
| Female | 1 | 0.0061 | 41.16 | 41.16 |
| | 2 | 0.0043 | 29.47 | 70.63 |
| | 3 | 0.0021 | 14.50 | 85.13 |
| | 4 | 0.0009 | 6.22 | 91.35 |
| | 5 | 0.0005 | 3.07 | 94.42 |
| | 6 | 0.0003 | 2.30 | 96.72 |
| | 7 | 0.0002 | 1.29 | 98.01 |
| Male | 1 | 0.0170 | 75.52 | 75.52 |
| | 2 | 0.0033 | 14.57 | 90.10 |
| | 3 | 0.0010 | 4.59 | 94.69 |
| | 4 | 0.0005 | 2.04 | 96.73 |
| | 5 | 0.0003 | 1.39 | 98.12 |

## GWA analyses

We performed GWA analyses using the PCs of natural variation in individual chromatographic peaks to identify novel components of the CHC metabolic pathways in *D. melanogaster*. None of the female or male PCs were affected by the presence of the endosymbiont *Wolbachia pipientis* in some of the DGRP lines (*Supplementary file 6*). Female PC1 and PC2 and male PC1 were affected by the *In(2L)t* inversion; female PC1, PC2 and PC3, and male PC5 were affected by the *In(3R)K* inversion; and the *In(3R)P* inversion only affected male PCs (PC1, PC3, PC5) (*Supplementary file 6*). We corrected for these effects prior to conducting the GWA analyses. Although these inversions contain genetic variants affecting CHC production, they cannot be resolved by GWA analysis due to elevated linkage disequilibrium within the inversions (*Mackay et al., 2012*; *Huang et al., 2014*) and are thus excluded from consideration.

We identified genetic variants in or near 305 (173) genes nominally ($P \leq 10^{-5}$) associated with female (male) PCs (*Supplementary file 7*). Although all of the top variants did not reach individual Bonferroni-corrected significance levels, most quantile-quantile plots indicated no systematic inflation of the test statistic and a clear departure from random expectation below $P < 10^{-5}$, justifying our choice of this reporting threshold and suggesting that the top associations were enriched for true positives (*Figure 6*).

Several of the top variants associated with each PC are in or near candidate genes with plausible roles in CHC metabolism (*Supplementary file 7*). In females, these include a single nucleotide polymorphism (SNP) in the fourth intron of *Lipase2 (Lip2)* associated with variation in PC1; a SNP 641 bp upstream of the cytochrome P450 gene, *Cyp49a1*, associated with PC2; SNPs in and near *Desiccate (Desi)*, a gene previously shown to contribute to desiccation resistance in *D. melanogaster*, associated with PC4 and PC6; variants in and near two peroxidase genes, *Immune-regulated catalase (Irc)* and *Peroxidase (Pxd)*, associated with PC5; and variants in three fatty acid metabolism genes (*CG14688, CG9458, CG8814*) associated with variation in PC6. Several notable candidate genes were also implicated by variants associated with PC7; *CG9801* contained the 3 most significant SNPs. One of the top associated SNPs causes a missense mutation in *Cyp6w1*. *Cyp4s3* and a predicted dihydrolipoamide branched chain acyltransferase (*St. Pierre et al., 2014*), *CG5599*, were down- and up-stream, respectively, of associated variants. We did not find any associated variants within or near *Desat1* or *Desat2*. In the case of *Desat2* the ancestral allele was not tested for association, because there were only six DGRP lines homozygous for the insertion, so its minor allele frequency (MAF = 6/169 = 0.035) did not reach the MAF cutoff (0.05) for evaluation. Further, any effect of this variant would have been minimized by correcting for the effect of the *In(3R)K* inversion.

In males, nearly all of the variants associated with variation in PC1 were in or near *CG13091*, a putative fatty acyl-CoA reductase, of which one, a nonsynonymous coding variant (*2L_8521314_SNP*), was the most significant variant in this study ($P$ = 2.19E-11) and passes the

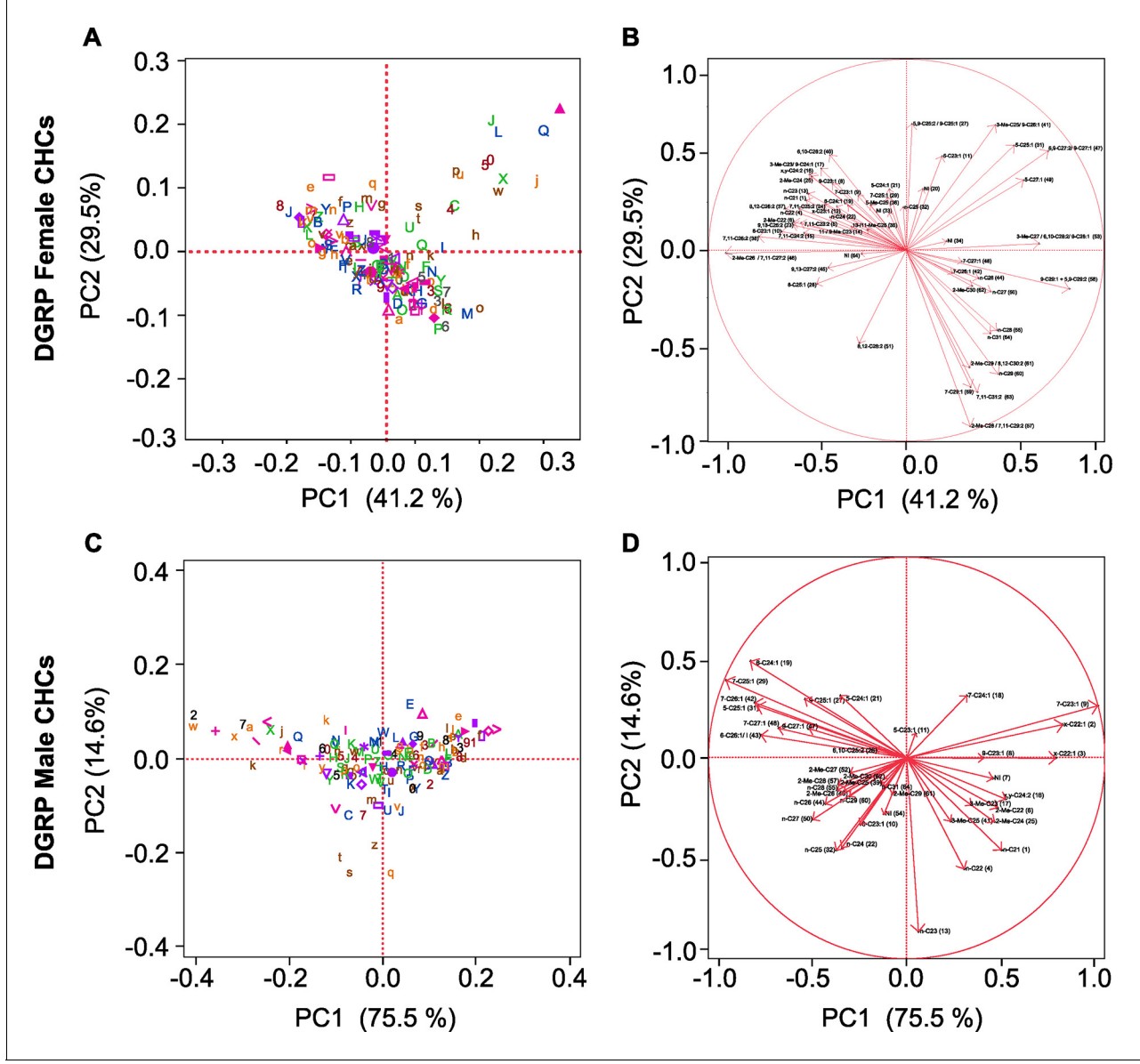

**Figure 5.** Principal component biplots for PC1 and PC2 of DGRP CHCs. (A) Female and (C) male PC1 and PC2. (B) Female and (D) male PC1 and PC2 eigenvectors. The percent of variance explained by each PC is indicated on the *x*- and *y*-axes. In (A) and (C) DGRP lines are color-coded (*Supplementary file 5*).

Bonferroni-correction for multiple tests (*Supplementary file 7*). Many of the variants in *CG13091* were in perfect or near-perfect linkage disequilibrium, and thus it is not possible to discern which was/were the causal variant(s). Variants in *CG13091* were also associated with variation in PC4 and PC5. Genes tagged by variants associated with PC2 include *approximated* (*app*), a palmitoyltransferase; *PHGPx*, a peroxidase; and *CG16979*, a thiolester hydrolase. Top variants in the PC3 GWA analysis included two genes predicted to be fatty acid elongases (*CG18609, CG30008*) and an NADH dehydrogenase (*CG8680*). The most significant variant in the PC5 GWA analysis was a nonsynonymous SNP (*3R_8220563_SNP*, P = 1.97E-09) in *CG10097*, which also encodes a fatty acyl-CoA reductase. Interestingly, there are two cytochrome P450 genes located upstream of *CG10097*, one functional (*Cyp9f2*) and one pseudogenized (*Cyp9f3Psi*).

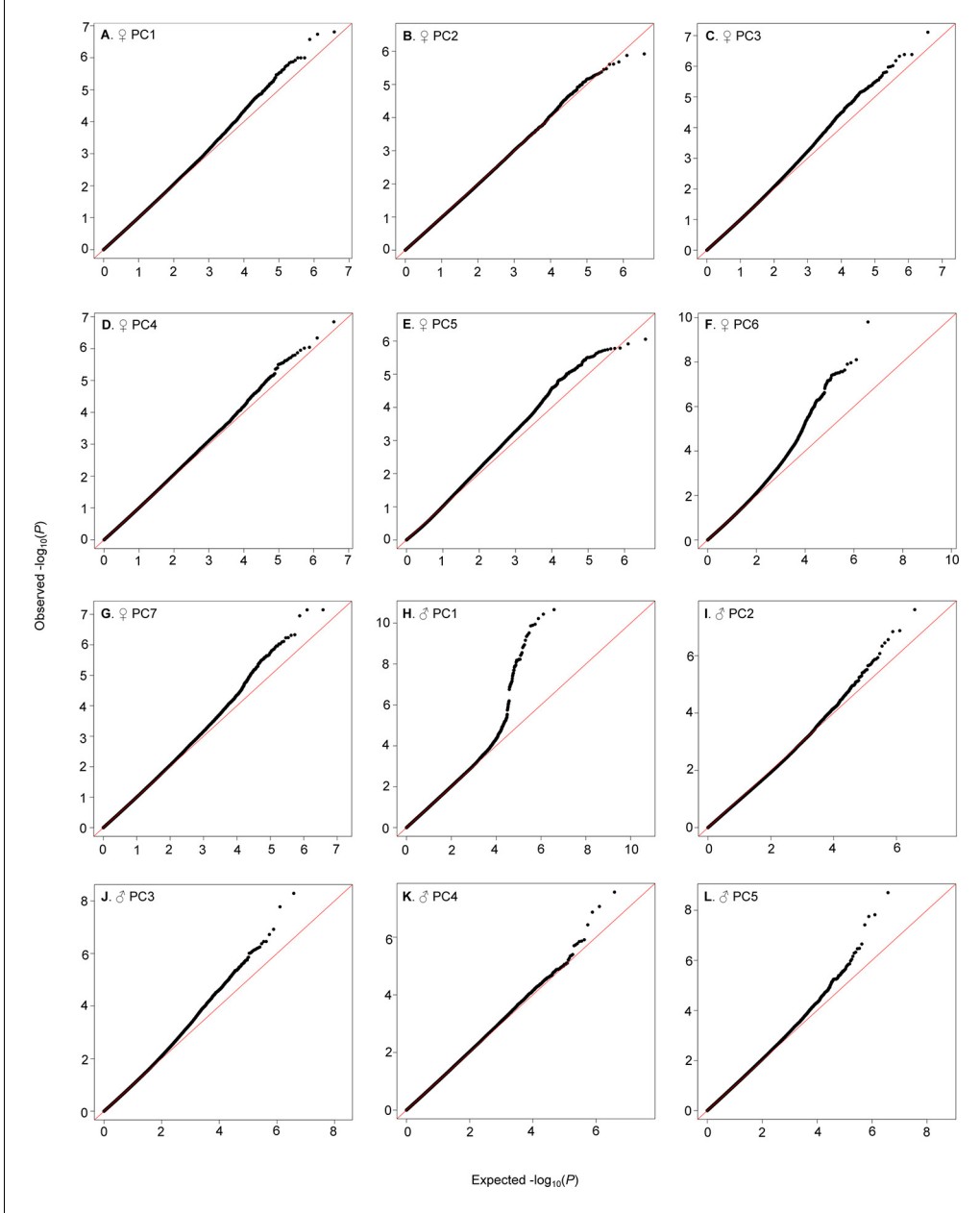

**Figure 6.** QQ-plots of CHC PCA GWA *P*-values. (**A–G**) Female PCs. (**H–L**) Male PCs.

## Functional validation of candidate genes

We selected 24 candidate genes with plausible contributions to various stages in the biosynthesis or turnover of CHCs and tested the effects of disruption of expression of these genes on CHC composition. All but one of these genes had publicly available transgenic *UAS*-RNAi lines (*Dietzl et al., 2007*). We used an oenocyte-specific *GAL4* driver line, *PromE(800)-GAL4* (*Billeter et al., 2009*), to restrict reduction in candidate gene expression to the CHC-producing cells. For the gene for which no RNAi line was available, *CG10097*, we used a *PiggyBac* insertion line to study the effects of this mutation on CHC composition (*Supplementary file 8*) (*Thibault et al., 2004*).

The candidate genes implicated by our GWA analyses that we selected for further functional assessment are annotated to encode a palmitoyl transferase (*app*), fatty acyl reductases (*CG13091* and *CG10097*), a thiol hydrolase (*CG16979*), thioredoxin peroxidases (*PHGPx* and *Prx6005*), fatty

acid elongases (*CG30008, CG18609,* and *CG9458*), cytochrome P450s (*Cyp49a1, Cyp9f2,* and *Cyp4s3*), peroxidases (*Irc, Pxd* and *Pxn*), an NADH dehydrogenase (*CG8680*), and a dihydrolipoamide branched chain acyltransferase (*CG5599*), which is involved in the metabolism of branched chain amino acids leucine, isoleucine and valine, which serve as precursors for methyl-branched alkanes (*Blomquist and Bagnères, 2010*). Disruption of expression of any of these genes by targeted RNA interference resulted in altered CHC compositions, often with sexually dimorphic effects. The alterations in CHC composition as a consequence of gene disruption were often complex and unexpected (*Figures 7* and *8*, *Figure 7—figure supplements 1–24*).

In insects, the evidence to date suggests that the elongated fatty acyl-CoA is reduced to an aldehyde prior to oxidative decarbonylation, the latter step being catalyzed by CYP4G1 in Drosophila (*Qiu et al., 2012*). While it is possible that the elongated fatty acyl-CoA is reduced from acyl-CoA to aldehyde to alcohol and then reoxidized to aldehyde before oxidative decarbonylation, to date there is no evidence for this. *CG13091* and *CG10097* encode fatty acyl-CoA reductases, which are both expressed at high levels in males and at low levels in females. FARs reduce fatty acyl-CoA to an alcohol (*Howard and Blomquist 2005*). Inhibition of the production of these FARs promoted the production of longer chain CHCs. Males in which *CG13091* expression was targeted with RNAi had higher relative amounts of longer-chain CHCs in general and reduced shorter-chain monoenes and methyl-branched CHCs. In particular, 7-C25:1 and 7-C27:1 and 2-MeC26, and 2-MeC28 increased substantially (*Figures 8* and *9*). The increase in 7-C25:1 is of particular interest, as this compound acts as a male sex pheromone that mediates female mate choice. Females showed a similar trend but to a lesser extent; they also had some increased longer-chain dienes and, like the males, increased 7-C27:1, 7,11-C27:2 & 2-MeC26, and 2-Me-C28 & 7,11-C29:2 (*Figures 7* and *9*). Both sexes of the *CG10097* mutant were similar, with higher relative amounts of longer-chain CHCs. They also had lower amounts of shorter-chain monoenes and methyl-branched CHCs. In the *CG10097* mutant and control females, we were able to separate the 7,11-C27:2 and 2-Me-C26 peaks as well as the 2-Me-C28 and 7,11-C29:2 peaks (*Figure 9*), enabling us to infer decreased 7,11-C27:2 and 7,11-C29:2 and increased 2-Me-C28 relative to the control. Our results suggest that the FARs encoded by *CG13091* and *CG10097* may be specifically associated with alkane and monoene synthesis.

The complexity of effects on CHC composition through RNAi-targeting is illustrated by the diverse effects on CHC composition of the three different fatty acid elongases encoded by *CG30008, CG18609* and *CG9458*. In males, *CG30008* may play a role in the elongation of precursors of *n*-alkanes and monoenes, and *CG18609* may elongate precursors of longer-chain *n*-alkanes and 2-methylalkanes (*Wicker-Thomas and Chertemps, 2010*). Disruption of *CG30008* expression in males resulted in greatly reduced amounts of total CHCs (~2-fold). The effect was less severe in females, but the overall trend was also towards reduced CHCs (*Figure 10*). Disruption of *CG18609* decreased longer-chain CHCs in both sexes but increased 2-Me-C24 in females (*Figure 7*) (*Wicker-Thomas and Chertemps, 2010*). Males also had increased total CHCs and large increases in shorter-chain CHCs (*Figure 8*). This effect was mimicked by interference with expression of *Cyp9f2*, which also resulted in fewer longer-chain monoenes, alkanes, and many methyl-branched CHCs. However, as for *CG18609*, 2-Me-C24 increased in females and other shorter-chain methyl-branched CHCs were trending upward. In males, there were overall increases, but especially in the shorter-chain CHCs. These observations suggest that *CG18609* may function in the biosynthesis of CHCs in coordination with *Cyp9f2*. Interestingly, the *CG9458* knockdown had sexually dimorphic effects on CHC production, increasing male CHCs and decreasing nearly all *n*-alkanes and monoenes in females. This suggests that *CG9458* is critical for the biosynthesis of *n*-alkanes and monoenes in females.

The three cytochrome P450s that we tested (*Cyp49a1, Cyp9f2, Cyp4s3*) all affected overall amounts of CHCs. RNAi knockdown of *Cyp49a1* and *Cyp9f2* in males led to general increases in CHCs, with a few exceptions. The increase in CHCs in males with compromised *Cyp49a1* function was accompanied by a decrease in many female CHCs. Reduced expression of *Cyp4s3* resulted in increased CHC abundance in both sexes. The link between these CYPs and CHC production and maintenance is unclear. However, we speculate that oxidation reactions mediated by these CYPs may regulate CHC degradation and turnover. The specific functions of most insect CYPs are still unknown (*Chung et al., 2009*). Given the role of CYP4G1 in CHC production (*Qiu et al., 2012*), we believe it is possible for these CYPs to have similar functions that are perhaps specific to particular subsets of CHCs.

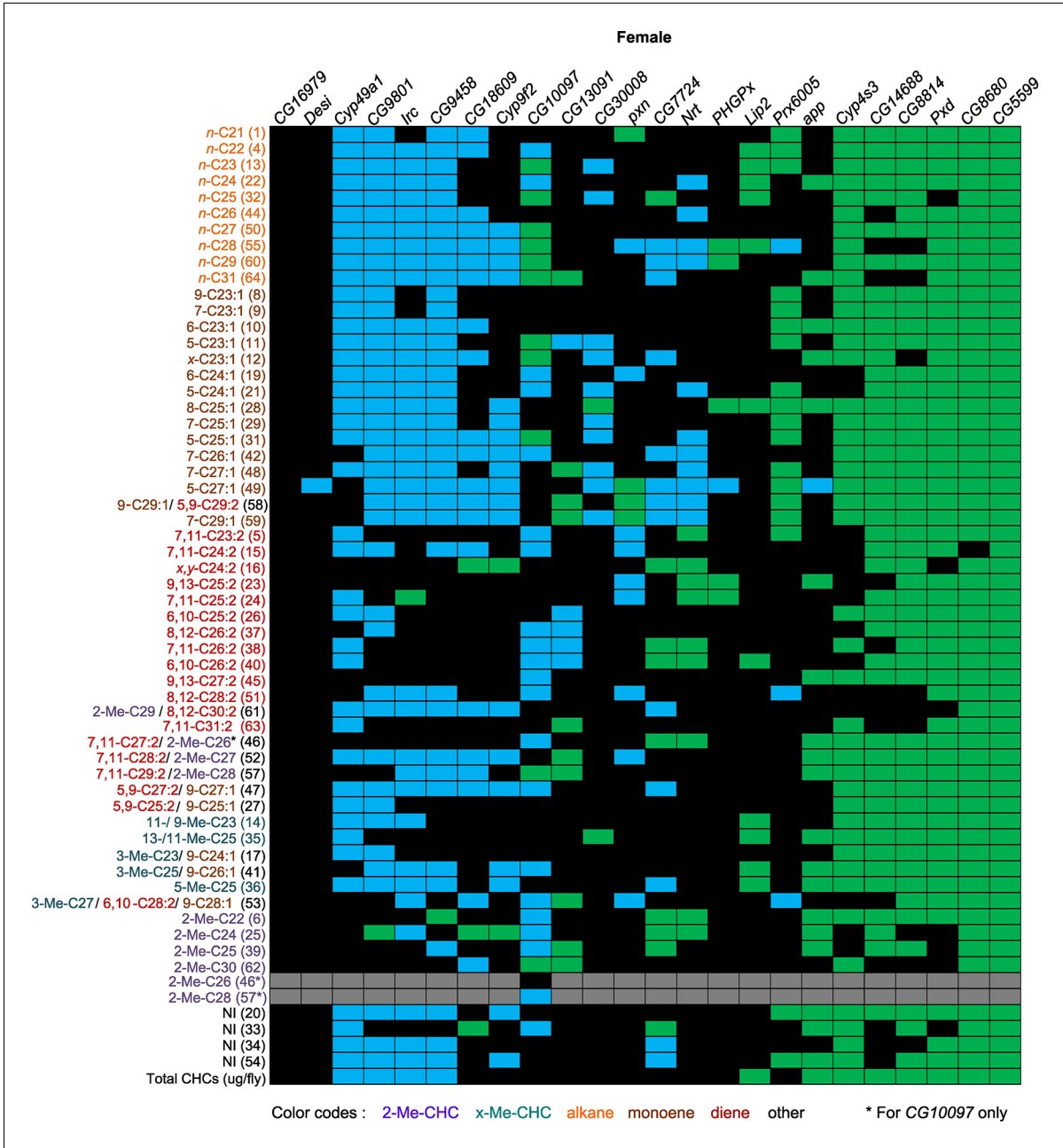

**Figure 7.** Summary of RNAi and mutant experiments for female CHCs. *UAS*-RNAi target gene and the *CG10097^{e00276}* mutant are indicated on the horizontal axis. CHC names and numbers are listed on the *y*-axis. Data are color coded to represent *P*-values ($P \leq 0.05$) from *t*-tests for the mean differences of the experimental and the control lines. Black = no significant change; blue = significant decrease; green = significant increase; gray = not applicable (peaks 46 and 57 split into two peaks for the *CG10097* mutant).

The following figure supplements are available for figure 7:

**Figure supplement 1.** Functional validation PCA and total CHCs for RNAi-*app*.

**Figure supplement 2.** Functional validation PCA and total CHCs for RNAi-*CG5599*.

**Figure supplement 3.** Functional validation PCA and total CHCs for RNAi-*CG7724*.

**Figure supplement 4.** Functional validation PCA and total CHCs for RNAi-*CG8680*.

*Figure 7 continued on next page*

*Figure 7 continued*

The complex interrelationships that give rise to variation in sexually dimorphic CHC profiles is further illustrated by RNAi interference of *Irc*, *Pxd*, and *Pxn*, which all have corresponding alleles associated with variation in CHC composition in the DGRP and encode peroxidases. However, interference with their expression through targeted RNAi resulted in different shifts in CHC composition. Disruption of *Irc* reduced the amounts of monoenes and increased the amounts of dienes in females, while increasing male CHCs, reminiscent of the effects of disruption of *Cyp49a1*. The effects of interference with *Pxd* were more complex. In females, nearly all CHCs increased, while in males many odd-chain CHCs increased, but there were decreases only in even-chain CHCs. A similar phenomenon was observed with disruption of *Cyp4s3*. *Pxn* may be important for diene synthesis because the knockdown females had decreased dienes and corresponding increased levels of longer-chain monoenes. However, in males there were decreased longer-chain monoenes, methyl-branched alkanes, and *n*-alkanes but increased shorter-chain CHCs. *CG7724* is inferred to contribute to oxidation-reduction processes and steroid synthesis. Disruption of *CG7724* expression in females resulted in a decrease in longer-chain monoenes and methyl-branched CHCs and an increase in 2-

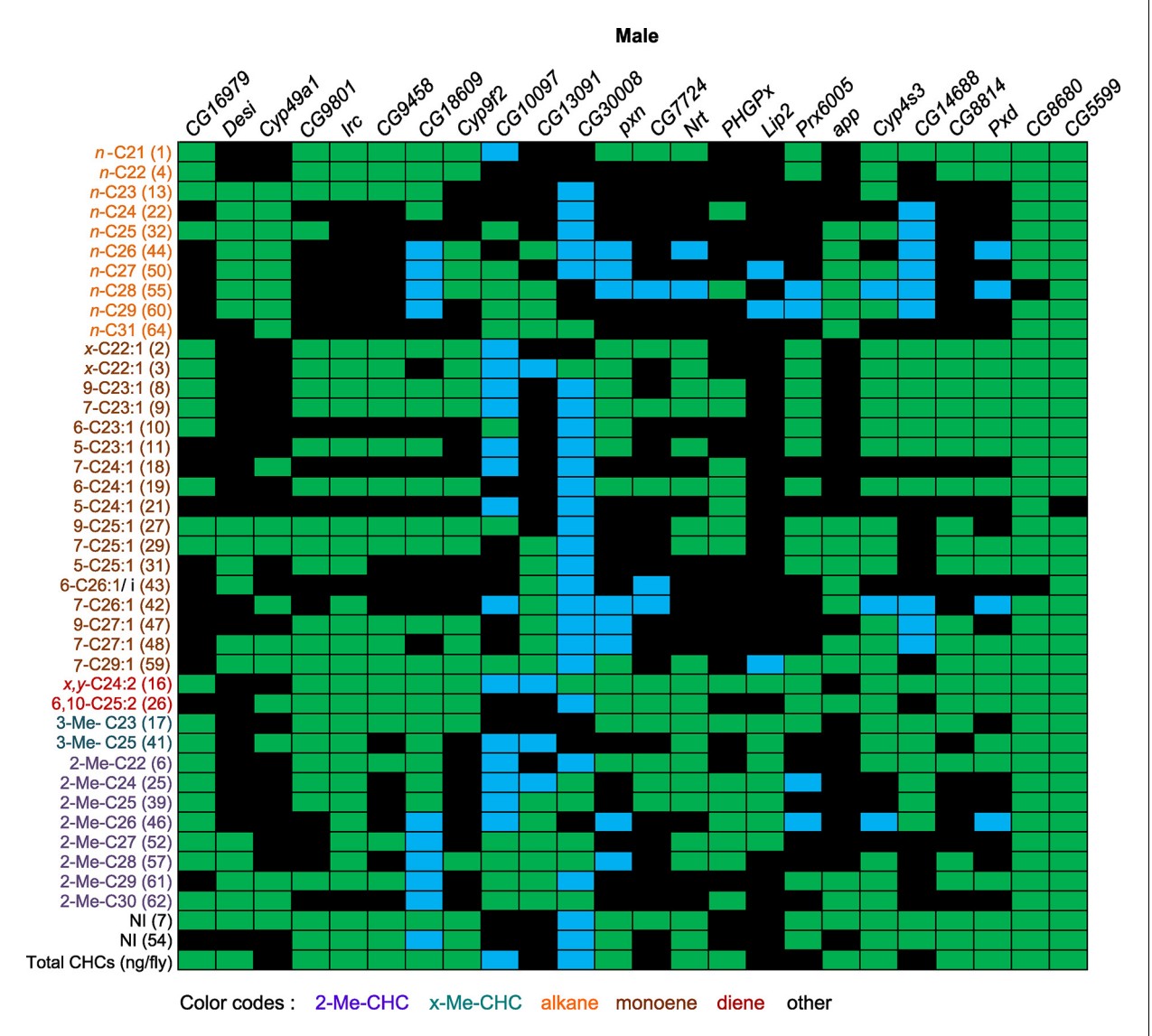

**Figure 8.** Summary of RNAi and mutant experiments for male CHCs. *UAS*-RNAi target gene and the *CG10097* [e00276] mutant are indicated on the horizontal axis. CHC names and numbers are listed on the *y*-axis. Data are color coded to represent *P*-values ($P \leq 0.05$) from *t*-tests for the mean differences of the experimental and the control lines. Black = no significant change; blue = significant decrease; green = significant increase; gray = not applicable (peaks 46 and 57 split into two peaks for the *CG10097* mutant).*

Me-C22, 2-Me-C24, 2-Me-C25, 2-Me-C26 and 7,11-C27:2, x,y-C24:2 and x,y-C26:2. Males also showed increases in 2-methyl alkanes but also in shorter-chain alkanes and monoenes. These results reveal a complex and dynamic network of oxidative enzymes of which the summed activity determines the sexually dimorphic composition of CHCs.

Disruption of the NADH dehydrogenase *CG8680* and of *CG5599*, which is predicted to have dihydrolipoamide branched chain acyltransferase activity, resulted in remarkable increases in the total amount of CHCs in both males and females (*Figures 7*, *8* and *10*, *Figure 7—figure supplements 2*, *4*, *Supplementary file 9*). Individual control female and male flies produced ~1.5–2 µg and ~1–1.5 µg of CHCs, respectively; in contrast, the RNAi-*CG8680* females and males produced ~5.5 µg and ~3 µg, respectively, of CHCs per fly (*Figure 7—figure supplement 4*). The *CG5599* knockdown caused a >4 µg increase per fly in each sex, representing ~3-fold (~7 µg/fly) and 4-fold (~5.5 µg/fly) increase for females and males, respectively. Thus, it is possible that these genes play roles in catabolic pathways thus resulting in an increase in CHCs. Alternatively, RNAi knockdown of these

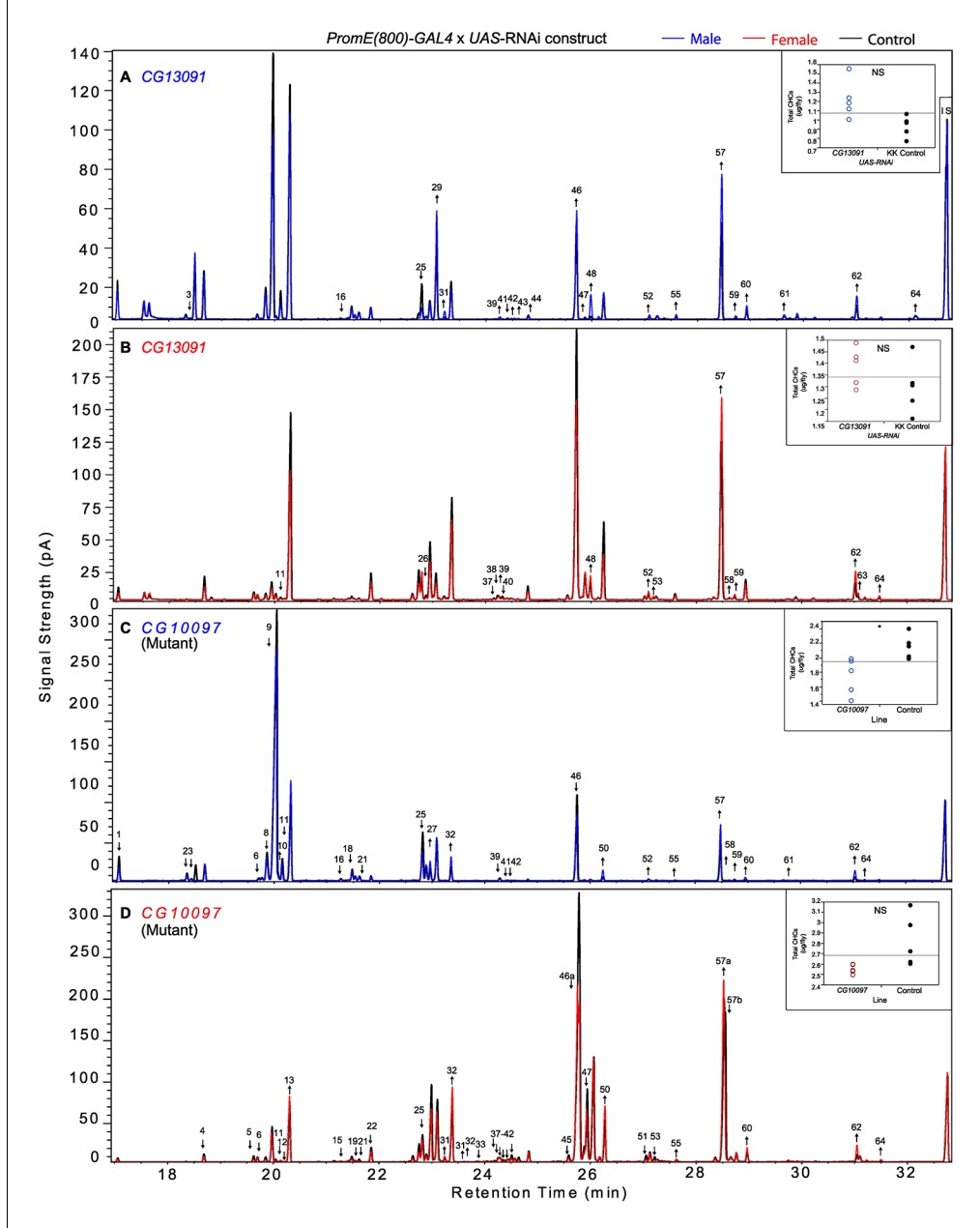

**Figure 9.** Example chromatograms of oenocyte-specific RNAi knockdowns and mutants – *CG13091* and *CG10097*. (**A**) and (**B**) *PromE(800)-GAL4* x *UAS-CG13091*. (**C**) and (**D**) Exelixis mutant *CG10097* [e00276]. pA = picoAmperes, IS = internal standard, ↑ CHCs significantly increased or ↓ decreased according to the individual t-tests.

genes may lead to reduced or abnormal grooming behavior leading to an accumulation of CHCs (*Böröczky et al., 2013*). Inhibition of expression of *Pxd*, *CG8814* and *Cyp4s3* also resulted in increased levels of total CHCs in both sexes, with notable decreases in 2-Me-C26 in *Pxd* and *Cyp4s3* males (*Figures 7* and *8*, *Figure 7—figure supplements 5*, *14*, *23*).

The gene products of *app*, *PHGPx*, and *Prx6005* may be involved in the release of fatty acids from the fatty acid synthase complex. The *app* palmitoyltransferase may be specific to methyl-branched fatty acids since we observe decreases in methyl-branched CHCs in males and females when its expression is disrupted. Male and female PCAs were here clearly separated from the controls and in both sexes the total amounts of CHCs were slightly increased (*Figures 7* and *8*, *Figure 7—figure supplement 1*). Additionally, males had elevated longer-chain methyl-branched

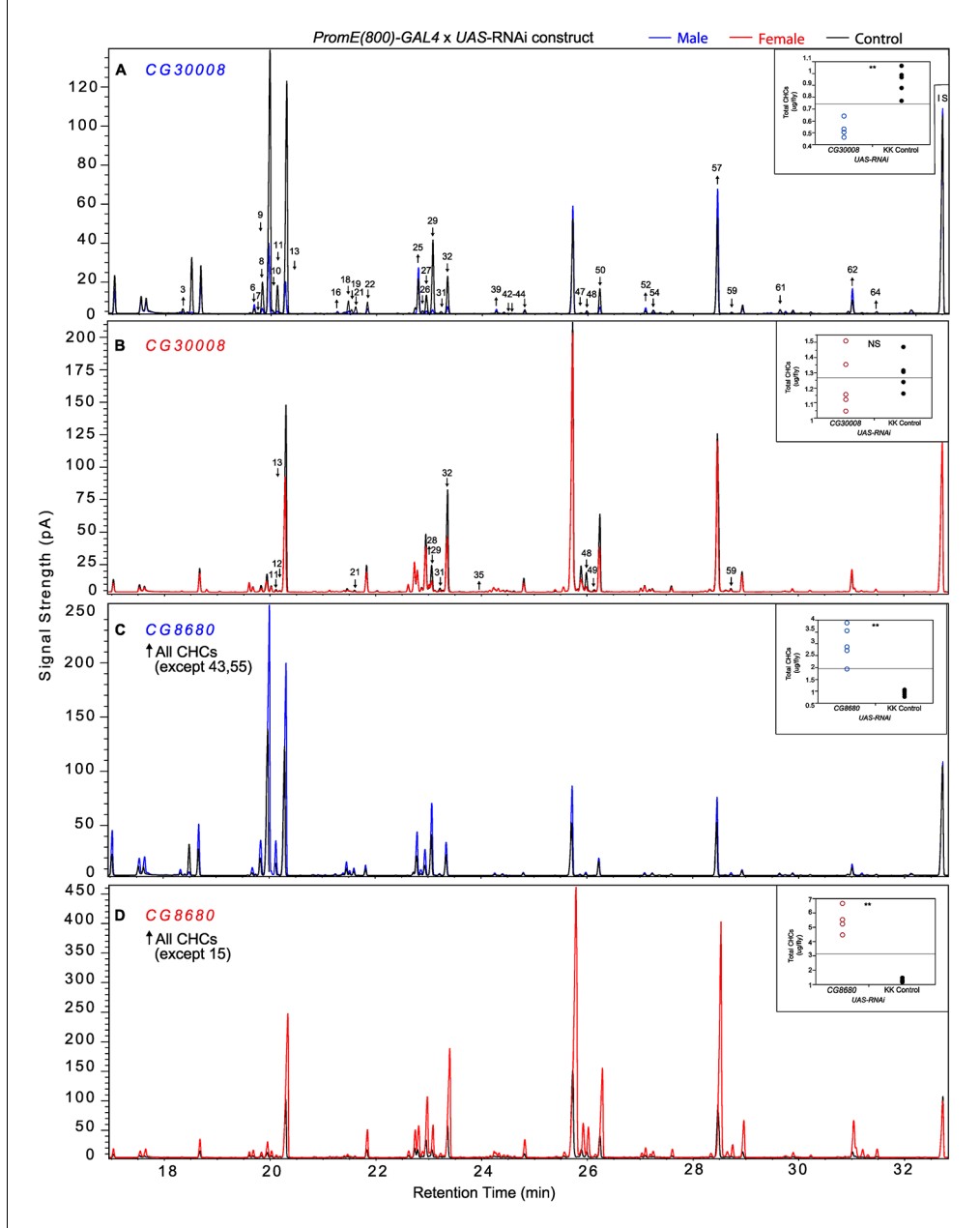

**Figure 10.** Example chromatograms of oenocyte-specific RNAi knockdowns – *CG8680* and *CG30008*. (A) and (B) *PromE(800)-GAL4* x *UAS-CG8680*. (C) and (D) *PromE(800)-GAL4* x *UAS-CG30008*. pA = picoAmperes, IS = internal standard, ↑ CHCs significantly increased or ↓ decreased according to the individual t-tests.

alkanes, *n*-alkanes and monoenes, while females had increased abundances of *n*-alkanes, methyl-branched alkanes, monoenes and dienes of both longer- and shorter-chain lengths. RNAi targeting of *Prx6005* increased shorter-chain monoenes and alkanes in both sexes. In females, longer-chain dienes and methyl-branched CHCs were trending downward, and in males levels of 2-Me-C26 and 2-Me-C24 decreased. Disruption of *PHGPx* increased the dienes 7,11-C25:2 and 9,13-C25:2, and some longer-chain *n*-alkanes in females. Levels of many short-chain monoenes and nearly all 2-methyl alkanes were elevated in males. Furthermore, disruption of *CG16979*, which encodes a gene product annotated as having thiolesterase activity, had no effect in females but caused many CHC increases in males, predominantly in monoenes. Total CHCs were also elevated (*Figures 7* and

*8*). Thus, there appears to be functional specialization among thiolesterases in the biosynthesis of CHCs.

We also assessed the effects of two candidate genes, *Desi* and *Lip2*, associated with variation in CHC composition in the DGRP that affect desiccation resistance. Expression of *Desi* fluctuates in wandering *D. melanogaster* larvae in response to environmental conditions and RNAi knockdown of *Desi* in larvae leads to higher mortality (*Kawano et al., 2010*). However, the nature of the resistance to desiccation is unknown and CHCs were not phenotyped in the larvae. In contrast to the GWA results, knocking down *Desi* had no effect on female CHCs, while the male CHCs clearly separated from the controls in the PCA (*Figure 7—figure supplement 17*). Males had increased alkanes, monoenes, and 2-Me-C28 (*Figure 8*). *Lip2* has been associated with clinal variation of life history traits in *D. melanogaster* populations in the eastern United States (*Fabian et al., 2012*), and has triglyceride lipase activity that could regulate release of free fatty acids for CHC synthesis. Females in which expression of *Lip2* was targeted with RNAi had increased *n*-alkanes and increased total CHCs, whereas the males had increased 2- and 3-methyl alkanes (*Figure 7*).

Finally, we examined the effects of *CG14688* and *CG9801*; little is known about the function of both of these genes (*St. Pierre et al., 2014*). Suppression of gene expression resulted in sexually dimorphic increases and decreases in overall CHCs. RNAi targeting of *CG14688* increased female CHCs but had more complex effects in the males, increasing shorter-chain *n*-alkanes and methyl-branched alkanes and decreasing longer-chain monoenes. Males expressing RNAi against *CG9801* had increased total CHC levels, while females of these lines had overall decreases in CHCs. In mammals, alpha-oxidation is used to chain-shorten 3-methyl-fatty acyl-CoAs by one carbon. We are not aware of any examples of alpha-oxidation in insects, but it could be involved in CHC metabolism.

## Discussion

Since CHCs represent the boundary between the organism and its environment and mediate social interactions while offering protection against adverse environmental effects, variation in CHC profiles may present a target for natural selection and adaptive evolution. Using the DGRP, we report one of the most comprehensive characterizations of natural variation in insect CHCs to date. We provide unambiguous evidence for extensive heritable individual variation in the relative abundances of CHC components. MMC analysis shows that CHC components are not independent, but can be organized as correlated modules, which likely reflect common biosynthetic origins. Variation in the CHC profiles can be captured with a limited number of PCs, which were used as composite phenotypic values for GWA analyses. This resulted in the identification of candidate genes associated with variation in CHC composition in males and females, including several genes which could be plausibly associated with CHC biosynthesis. Despite a lenient significance threshold for reporting associations, targeted gene disruption of each of these 24 candidate genes indeed affected CHC composition, often in a sexually dimorphic manner. Our results highlight the complex interactions among genes that directly affect the composition and accumulation of CHCs and regulatory elements that affect CHC indirectly through general lipid metabolism and nutrition (see also *Wicker-Thomas et al., 2015*).

### Identification of novel CHC components

We found substantial heritable natural variation in CHC composition for males and females of the inbred, sequenced DGRP lines. Several of the epicuticular compounds identified in this study have not been reported previously for *D. melanogaster* (*Jallon and David, 1987*; *Foley et al., 2007*; *Everaerts et al., 2010*; *Dweck et al., 2015*). These compounds separated in our GC analyses because we used a thin high-resolution column and a relatively long temperature program. Several of the newly identified monoenes and dienes had double bonds in an even carbon position, an unusual configuration in insects. While most of these new compounds represented a very small fraction of the total CHCs, two (peak 51 = 8,12-C28:2 and peak 53 = 6,10-C28:2, 9-C28:1 and 3-Me-C27) were female-specific and could potentially play a role in sexual communication.

### A segregating African CHC phenotype in a Cosmopolitan population

The African CHC phenotype has an abundance of 5,9-C27:2 and lower levels of 7,11-C27:2, and is present only in populations from Sub-Saharan Africa and the Caribbean (*Coyne et al., 1999*). Based

on genotyping the 16-bp deletion/ancestral *Desat2* allele, Caribbean populations are known to have spread northward into the southern United States; however, populations north of Alabama and Mississippi were thought to be nearly fixed for the Cosmopolitan deletion (*Yukilevich and True, 2008*). Surprisingly, we found that the DGRP is segregating for the African CHC phenotype. Thus, to the best of our knowledge the DGRP progenitor population at the North Carolina Farmers Market represents the northern most population of *D. melanogaster* with the African *Desat2* allele documented to date.

While 17 DGRP lines contain the ancestral allele that confers a functional *Desat2*, only 15 of these lines exhibited the African phenotype. On average, females of DGRP lines that were heterozygous or homozygous for the functional allele, regardless of inversion status, had intermediate amounts of the sex pheromone CHC peaks relative to the African lines or DGRP lines with the deletion. However, one DGRP line, DGRP_367, had more 5,9-C27:2 than either African line. Two DGRP lines (DGRP_105, ins/ins, INV/ INV and DGRP_551, ins/del, ST/ ST) had the functional *Desat2* allele, yet exhibited the Cosmopolitan phenotype. These results are in contrast to previous reports of complete association of the ancestral allele with the African female CHC phenotype (*Dallerac et al., 2000*; *Takahashi et al., 2001*). Further analyses showed that other *Desat2* polymorphisms in DGRP_105 likely result in a non-functional protein. However, the other Cosmopolitan-like line, DGRP_551, is puzzling because it is heterozygous for the functional allele, but homozygous for the standard karyotype. It is possible that this line is segregating for the *In(3R)K* inversion at low frequency and individuals with the ST/INV karyotype were not detected, but the lack of correspondence between the functional allele and CHC status remains unexplained since *Desat2* itself does not harbor a potentially damaging mutation in this line. Perhaps a polymorphism in an unknown gene in DGRP_551 interacts with the *Desat2* functional allele to suppress its effects. Finally, there were four DGRP lines that exhibited the African phenotype and were homozygous for the ancestral allele, but they were ST/ST in karyotype. One possible explanation for this 'mismatching' of phenotype and genotype with the karyotype is that the *Desat2* 16-bp deletion occurred prior to the inversion event. This would mean that the inversion may be segregating for the *Desat2* 16-bp allele.

## GWA analysis and functional assessment of candidate genes reveals a complex enzymatic network for CHC biosynthesis and turnover

We used principal component analysis to reduce dimensionality of the data, which was motivated by the high co-linearity among the CHCs. However, PCA may cause genetic variation for a certain CHC to be distributed among multiple PCs and therefore dilute its association with QTLs. Surprisingly, we did not detect variants in the DGRP in previously identified CHC biosynthesis genes (*FASN1, Desat1, eloF, DesatF, Cyp4G1*; and several genes reported in *Wicker-Thomas et al., 2015*) associated with CHC variation. However, we did find many novel candidate genes, and functional tests showed that disruption of expression of all tested candidate genes had significant effects on the amount of CHC on the cuticular surface. While the mechanistic relationships between any of these genes are unknown, some share commonalities in their phenotypes when their expression is disrupted with RNAi. However, the majority of genes which encode gene products with similar molecular functions result in different shifts in CHC profiles when disrupted. Thus, variation in the CHC profile arises as an emergent phenotype from the dynamics of complex interrelated biosynthetic and catabolic pathways.

The RNAi-induced shifts in CHC profiles are frequently sexually dimorphic. This could reflect different expression levels of metabolic enzymes associated with CHC production. However, we cannot exclude the possibility that differential effectiveness of RNAi in males and females may contribute to apparent sexual dimorphism. In addition, we note that phenotypic changes associated with knocking down expression of a target gene with RNAi may not be causal, but a consequence of off-target effects of RNAi or the *GAL4* driver. Further studies are needed to clarify the effects on CHC production of these genes and their interactions, and to test specific mechanisms and enzymatic activities through which they exert these effects. Furthermore, it is important to note that the RNAi and mutant experiments test for effects at the level of genes and are only proxies for the effects of segregating natural variants in these genes.

We recognize that the 24 candidate genes on which we focused represent only a subset of all candidate genes associated with variation in CHC composition. Many of these genes may directly or indirectly affect CHC composition through as yet unknown mechanisms. GWA studies of

glucosinolates in *Arabidopsis thaliana* (*Chan et al., 2011*), flowering time in maize (*Buckler et al., 2009*), human height (*Yang et al., 2010*), and other traits in *D. melanogaster* such as body pigmentation (*Dembeck et al., 2015*) have also shown that the genetic architecture underlying variation in potentially adaptive traits includes many polymorphic loci with small effect sizes. Thus there is no reason to assume that variation in adaptive traits is controlled by few, large effect loci. Our results provide a framework for future studies of the mechanisms that regulate CHC composition and their adaptive potential regarding cold/heat tolerance and desiccation resistance, and pleiotropic effects on chemical communication and mate choice.

## Materials and methods

### *Drosophila* stocks and phenotyping

DGRP and African (Z30 and Z53) lines were reared in vials containing cornmeal-molasses-agar medium at 25°C, 75% relative humidity, a 12:12-h light-dark cycle, and a controlled adult density of 10 males and 10 females. The parental generation was allowed to lay eggs for three days. Upon eclosion, virgin males and females were separated and placed into new vials containing the same medium and aged for four days.

The flies from each line were separated into at least two samples of five flies each per sex. On average, three samples were collected for each. To avoid cross-contamination of cuticular lipids a fresh tissue paper was placed on the carbon dioxide pad and the flies were handled with acetone-washed titanium forceps at each round of sorting. Flies were placed in 2 mL glass auto-injection vials with a Teflon cap and were flash frozen. All samples were stored at -30°C until CHC extraction. We collected samples from 169 and 157 DGRP lines for females and males, respectively (1,078 total samples). All lines were reared simultaneously and DGRP lines that did not produce sufficient offspring for CHC analysis were excluded to avoid any block effects of rearing. For the two African lines, Z30 and Z53, we reared and collected 5 samples for each sex.

### Quantification and identification of cuticular hydrocarbons

Cuticular lipids were extracted from each sample using 200 µl of hexane containing an internal standard (IS, 1 µg *n*-C32) with gentle swirling for five minutes. The flies were briefly extracted a second time with 100 µl of hexane (free of internal standard). After each wash the extract was transferred to a 300 µl conical glass insert. The extract was dried using a gentle stream of high-purity $N_2$ and resuspended in 50 µl of hexane. The samples were immediately processed using gas chromatography or stored at 4°C (no longer than one day) until processing.

The cuticular lipid extracts were analyzed using an Agilent 7890A gas chromatograph with a DB-5 Agilent capillary column (20 m x 0.18 mm x 0.18 µm) and a flame ionization detector (FID) for quantification. We introduced 1 µl of sample using an Agilent 7683B auto injector into a 290°C inlet operated in splitless mode. The split valve was turned on after 1 min. The oven temperature program was as follows: 50°C for one min, increased at 20°C/min to 150°C, and increased at 5°C/min to 300°C followed by a 10 min hold. Hydrogen was used as the carrier gas at constant flow (average linear velocity = 35 cm/sec) and the FID was set at 300°C.

Selected samples were analyzed for chemical identification in a 6890N GC system (Agilent) coupled with a 5975 mass selective detector (MSD) (Agilent) and equipped with a DB-5 (20 m x 0.18 mm x 0.18 µm) column (Agilent). Helium was used as carrier gas at 33 cm/s average linear velocity. Injection and temperature settings were identical to the settings described above, and the transfer line was maintained at 300°C. Positive electron ionization at 70 eV with default temperature settings (ion source at 150°C, quadrupole at 230°C) were used for the MSD. Ions were detected in scan mode in the range of 33–650 *m/z* at 1.23 scan/s scan rate. Compounds were identified based on their mass spectra in comparison to those in the reference library (Wiley 7th/NIST 05) and based on comparison of their retention indices and fragmentation patterns to already published *Drosophila* CHCs (*Howard et al., 2003*; *Everaerts et al., 2010*; *Dweck et al., 2015*).The position of the double bonds was not confirmed by performing microderivatization reactions and chirality was not determined for any of the CHCs. More details on how the putative structure of previously unpublished *D. melanogaster* CHCs was deduced (based on mass spectra and retention index data) are given in *Supplementary file 10*.

All chromatograms were analyzed using Agilent ChemStation software. For quantification individual peak areas were obtained for 42 and 60 male and female CHC containing peaks, respectively (some of the peaks contained multiple CHCs). Response factors were not determined for individual components. To account for natural variation in body size and absolute amounts of CHCs between the lines, the data were represented as proportions by dividing each peak area by the sum of all integrated peaks.

## Statistical and quantitative genetic analyses

We partitioned variation of each CHC peak into genetic and environmental components using an ANOVA model of form $Y = \mu + L + \varepsilon$, where $Y$ is phenotype, $\mu$ is the overall mean, $L$ is the random effect of line, and $\varepsilon$ is the residual. We estimated variance components using restricted maximum likelihood and computed the broad-sense heritability ($H^2$) of each CHC peak as $H^2 = \sigma^2_L/(\sigma^2_L + \sigma^2_\varepsilon)$, where $\sigma^2_L$ is the among-line variance component and $\sigma^2_\varepsilon$ is the error variance. All analyses were performed with version 9.3 of the SAS System for Windows (2013 SAS Institute Inc.).

A majority of CHCs belong to homologous series in which the chain length increases by two carbons; thus these compounds may be genetically correlated due to shared biosynthetic pathways and the data may be confounded with multi-co-linearity (*Martin and Drijfhout, 2009*). We visualized the correlations between CHCs using modulated modularity clustering (MMC) (*Stone and Ayroles, 2009*). The MMC algorithm clusters highly correlated variables based on the Spearman's rank correlation coefficients (ρ). In order to take these correlations into account, we conducted principal components (PC) analysis on the variance-covariance matrices for the male and female CHC line means. For each analysis we included only CHC peaks that had an estimated $H^2 \geq 0.25$. We retained PCs explaining greater than 1% of the variation for subsequent GWA analysis. PCA was conducted in JMP Pro10 (2013 SAS Institute Inc.).

## Genome-wide association (GWA) analyses

We conducted a GWA analysis for each CHC PC, separately for males and females. The DGRP lines are segregating for Wolbachia infection status and for the following common inversions: *In(2L)t, In(2R)NS, In(3R)P, In(3R)K*, and *In(3R)Mo*. We performed GWA studies in two stages. In the first stage, we adjusted the line means for the effects of Wolbachia infection and major inversions. We then used the adjusted line means to fit a linear mixed model in the form of $Y = \mathbf{X}b + \mathbf{Z}u + \varepsilon$, where $Y$ is the adjusted phenotypic value, $\mathbf{X}$ is the design matrix for the fixed SNP effect $b$, $\mathbf{Z}$ is the incidence matrix for the random polygenic effect $u$, and $\varepsilon$ is the residual. The vector of polygenic effects $u$ has a covariance matrix in the form of $\mathbf{A}\sigma^2$, where $\sigma^2$ is the polygenic variance component. We fitted this linear mixed model using the FastLMM program (version 1.09) (*Lippert et al., 2011*). We performed these single marker analyses for the 1,883,938 (females) and 1,912,894 (males) biallelic variants (SNPs and indels) with minor allele frequencies $\geq 0.05$ whose Phred scale quality scores were at least 500 and genotypes whose sequencing depths were at least one and genotype quality scores at least 20 (*Huang et al., 2014*). All segregating sites within lines were treated as missing data.

## RNAi and mutant candidate gene validation experiments

We selected candidate genes with available mutations and RNAi knockdown constructs to test for effects on CHC production based on FlyBase annotations. We obtained lines with RNAi knockdown constructs from the Vienna *Drosophila* RNAi Center (VDRC) and crossed them to the oenocyte-specific *GAL4* driver, *PromE(800)-GAL4* (*Dietzl et al., 2007*; *Billeter et al., 2009*). We tested knockdown constructs and their co-isogenic controls (F1 individuals from crosses of the empty vector strain to *PromE(800)-GAL4* for 23 genes (*Supplementary file 8*). Since no RNAi knockdown line was available from VDRC for *CG10097* identified in the male GWA analysis, we obtained and tested for this gene a *PiggyBac* insertion mutant from the Harvard Exelixis Collection (*Thibault et al., 2004*) along with the *w1118* control line. From each cross and mutant line, we collected and aged both male and female virgins and analyzed the CHCs in the same manner as described for the DGRP flies. The analysis of CHCs using GC was also the same. However, for these lines, instead of calculating the proportion that each peak contributed to the total chromatogram, we used the internal standard to calculate the amount of each CHC present in the sample (ng/fly) with the assumption that body sizes between the control and RNAi knockdown or mutant were not significantly different. These

measures provide a more quantitative measure of CHCs and capture differences that proportion data may not resolve. PC analyses and *t*-tests pairing the test lines with the controls were conducted on these data. We also calculated the mean total amount of CHCs (ng/fly); we used the more conservative Cochran and Cox test, which assumes unequal group variances to assess statistically significant effects on CHC composition. We also present these data as µg/fly in *Figure 7—figure supplement 1–24*. *t*-tests were conducted in SAS v. 9.4 (2013 SAS Institute Inc.).

## Acknowledgements

We thank Rick Santangelo for technical support. We also thank Drs. Akihiko Yamamoto and Richard Lyman for maintenance of fly lines and Drs. Joel Levine and Pat Estes for sharing the *PromE(800)-GAL4* line. We acknowledge the Vienna *Drosophila* RNAi Center and the Exelixis Collection at the Harvard Medical School.

## Additional information

### Funding

| Funder | Grant reference number | Author |
|---|---|---|
| National Institutes of Health | R01 GM59469 | Robert R H Anholt<br>Trudy F C Mackay |
| Blanton J Whitmore Endowment | | Coby Schal |
| National Institutes of Health | R01 GM45146 | Robert R H Anholt<br>Trudy F C Mackay |

The funders had no role in study design, data collection and interpretation, or the decision to submit the work for publication.

### Author contributions

LMD, KB, WH, CS, RRHA, TFCM, Conception and design, Acquisition of data, Analysis and interpretation of data, Drafting or revising the article

## Additional files

### Supplementary files

• Supplementary file 1. DGRP line means and African samples CHC data. (A) DGRP females. (B) DGRP males. (C) DGRP and African females. (D) DGRP and African males.

• Supplementary file 2. ANOVA and heritabilities of DGRP CHC peak proportions in females and males. (A) Females. (B) Males. df: degrees of freedom; SS: Sums of squares (Type III); MS: Mean squares; F: F ratio test statistic; *P*: *P*-value; $\sigma^2$: variance component; $H^2$: broad sense heritability.

• Supplementary file 3. ANOVA of female sex pheromones in DGRP lines containing the *Desat2* insertion (ins/ins or ins/del). df: degrees of freedom; *P*: *P*-value.

• Supplementary file 4. CHC module correlations in the DGRP. (A) Females. (B) Males.

• Supplementary file 5. Color and symbol codes for DGRP lines used in *Figure 5*.

• Supplementary file 6. ANOVA of the effects of *Wolbachia* infection and common polymorphic inversions on CHC PCs. df: degrees of freedom; SS: Type III sums of squares; F: F statistic; AIC: Akaike information criterion. ***p<0.001; **p<0.01; *p<0.05.

• Supplementary file 7. GWA results for DGRP female and male CHC PCs. Results are given for female PC 1 - 7 and male PC 1 - 5. *P*-values ($\leq 10^{-5}$) of association tests are given for a linear mixed model accounting for relatedness of adjusted line mean PCs versus genotypes.

• Supplementary file 8. VDRC and Exelixis transgenic line information.

• Supplementary file 9. Difference between experimental lines and controls for individual CHCs (ng/fly) and raw data (μg/fly). (A) Mean difference between RNAi or mutant CHCs and controls, females. (B) Mean difference between RNAi or mutant CHCs and controls, males. (C) Experiment 1 raw data for KK RNAi lines, females. (D) Experiment 2 raw data for KK RNAi lines, females. (E) Raw data for GD RNAi lines, females. (F) Raw data for Exelixis mutation, females. (G) Experiment 1 raw data for KK RNAi lines, males. (H) Experiment 2 raw data for KK RNAi lines, males. (I) Raw data for GD RNAi lines, males. (J) Raw data for Exelixis mutation, males.

• Supplementary file 10. Chemical identification of previously unpublished CHCs in *Drosophila melanogaster*.

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
