## [Decision Letter]

Thank you for submitting your work entitled "Genetic architecture of natural variation in cuticular hydrocarbon composition in *Drosophila melanogaster*" for peer review at *eLife*. Your submission has been favorably evaluated by Detlef Weigel (Senior editor) and three reviewers, one of whom, Daniel J Kliebenstein, is a member of our Board of Reviewing Editors.

The reviewers have discussed the reviews with one another and the Reviewing editor has drafted this decision to help you prepare a revised submission.

We felt that the manuscript was very interesting but that some key factors need amendment to reflect insect biochemistry and editorial changes to increase the potential audience of this nice genetic analysis. We therefore ask you to do the following:

1) Rework the manuscript to properly reflect the specifics of FA biosynthesis in insects rather than rely upon the plant model of the pathway.

2) Strive to present the broader picture and impacts of the results found in this study beyond CHC metabolism in *Drosophila*.

3) Clarify that the candidate genes found may influence the CHC accumulation but further work needs to be done to test the specific enzymatic activities.

4) Discuss issues on chemical identification and quantification.

Explicit reviews can be found below.

*Reviewer #1:*

The authors investigate the quantitative genetic variation in CHC hydrocarbons within *Drosophila* using a GWA approach and identify a highly polygenic underpinning to this trait. They then validate the potential role of 24 candidate genes in altering this trait. This analysis was very nicely conducted but the writing was solely built around CHC composition in *Drosophila* which will make it difficult for this paper to attract the potential readership that it should obtain. From my reading, there are two broad messages that all researchers using GWA should be able to obtain from this manuscript.

1) It is possible to being dissecting highly polygenic GWA traits that have 100s or more potential candidate genes down to the functional level. There is no need to assume that there are few causal genes for potentially adaptive traits.

2) Polygenic adaptive traits may be structured in a manner that will allow data decomposition approaches to allow significant progress in understanding complex traits like metabolic mixtures.

The manuscript would likely get broader visibility if there was more general information about the bigger picture presented in the manuscript that could pull in researchers studying GWA or metabolic traits in other systems as this manuscript has information for those researchers even if they are studying plants or fungi or humans. While motivated researchers will see the broader impacts of this manuscript on the GWA/metabolite field it would help to work and bring them into the research without this effort by presenting the broader ramifications of the research in the Introduction and Discussion.

While I appreciate the use of PCs to reduce the dimensionality, this is an inherently linear approach that can have trouble with non-linear processes such as epistatic variation in biosynthetic pathways. This can be illustrated by Figure 5 where the PC1 x PC2 plot shows two linear relationships. One going NW to SE and the other from NE to SW, this suggests that PC1 and PC2 are capturing variation at potentially biosynthetic loci that work within the pathway to control structural variation. Did the authors attempt to parse out these potential biosynthetic clues to synthesize a pathway? Another reason to consider this is that the PC approach can hide key biosynthetic genes at the intersection of the two PCs. This is true in the glucosinolate work within *Arabidopsis* wherein the PC approach utilized by Bergelson and colleagues was unable to identify all of the key biosynthetic enzymes that could be identified on a more metabolite by specific metabolite analysis or alternatively using ratios of precursors to products (Wentzell et al., 2007; Chan et al., 2011; Brachi et al., 2015). While I don't feel this is necessary to conduct, it would be beneficial to discuss this difficulty that is imparted by the use of linear trait decomposition for GWA.

Brachi, B. et al.(2015). Coselected genes determine adaptive variation in herbivore resistance throughout the native range of *Arabidopsis thaliana*. Proceedings of the National Academy of Sciences of the United States of America 112, 4032-4037.

Chan, E.K. et al. (2011). Combining genome-wide association mapping and transcriptional networks to identify novel genes controlling glucosinolates in *Arabidopsis thaliana*. PLoS Biol 9, e1001125.

Wentzell, A.M. et al. (2007). Linking metabolic QTLs with network and cis-eQTLs controlling biosynthetic pathways. PLoS Genet 3, 1687-1701.

*Reviewer #2:*

This manuscript represents a great deal of very good work and can be made into an excellent publication. However, much of the biochemistry upon which it is based needs extensive correcting and rewriting.

The identification of new components in this much studied species is well done and important.

In the subsection “Functional validation of candidate genes”, second paragraph: Fatty acid biosynthesis begins with an acetyl-CoA and then elongation occurs with a malonyl-CoA in which the energy is supplies by decarboxylation of the malonyl-CoA as two carbons incorporated into the growing chain. The next three steps are a reduction of a carbonyl to an alcohol, dehydration, and reduction of the carbon-carbon double bond. The authors are making too extensive use of plant lipid metabolism data. The FAS in animals, including flies, occurs on the multifunctional protein fatty acid synthetase (FAS) in which the growing chain is attached to an acyl carrier type unit that is part of the multifunctional (7 enzyme activities on one protein, act in a dimer). It is not typically called an acyl carrier protein complex. Products of insect FASs, particularly *Drosophila* FAS, can be 14, 16 or 18 carbons. A thioesterase that is part of the multienzyme FAS removes the elongated chain as a free fatty acid. I do not know what is meant by conjugating and the acyl group is never removed by a thioreductase. In all cases studied in insects, desaturation and elongation take place in a microsomal fraction (endoplasmic reticulum) and not the mitochondrial fraction (many references).

In the subsection “Functional validation of candidate genes”, fourth paragraph: Fatty acids are elongated and desaturated attached to an ACP in plants. In insects, the parallel reactions occur with the acyl group attached to CoASH. The evidence to date suggests that the elongated fatty acyl-CoA is reduced to an aldehyde prior to oxidative decarbonylation, the latter step catalyzed by *Cyp4G1* in *Drosophila* (Qiu et al., 2012). While it is possible that the elongated fatty acyl-CoA is reduced from acyl-CoA to aldehyde to alcohol and then re-oxidized to aldehyde before oxidative decarbonylation, there is no evidence for this to date in insects. (See Qiu et al. and the Reed et al. papers referenced there.)

In the subsection “Functional validation of candidate genes”, end of fourth paragraph: An esterase cleaves an ester bond and does not reduce the carboxylic ester group on the fatty acyl-CoA (not ACP) to an alcohol. A FAR (fatty acyl-CoA reductase) does the reduction.

In the subsection “Functional validation of candidate genes”, sixth paragraph: I am not convinced the CYPs studied here are involved in CHC turnover.

In the subsection “Functional validation of candidate genes”, last paragraph: The authors are comparing 3-methyl fatty acids with a 3-methyl hydrocarbon derived from a n-3 methyl fatty acid.

When stating “Our results provide a new perspective and highlight the complexity of the biosynthetic and catabolic pathways that regulate the dynamics of CHC composition”, the authors are blithely referring to the biosynthetic pathways of hydrocarbons (which are somewhat known) with the catabolic pathways (which to my knowledge almost nothing is known). This sentence is illustrative of several times in the manuscript where the authors make sweeping statements of biochemistry for which there is no basis.

*Reviewer #3:*

This manuscript by Dembeck et al. is the latest in a series of fine papers from the Mackay lab using the sequenced inbred lines from the DGRP to understand the genetic architecture of phenotypic trait variation, in this case, the variation in cuticular hydrocarbons (CHC) in *Drosophila melanogaster*. This manuscript reports three main findings: 1) the discovery of 16 previously unidentified CHCs in *D. melanogaster*, 2) that reported *Desat2* alleles do not match reported female CHC phenotypes; and 3) the identification of previously unidentified candidate genes in the biosynthesis of CHCs (using genome-wide association analysis to identify candidates and their functional validation by either RNAi knockdown or transposon insertion).

The main strengths of this manuscript are its systematic approach to gene identification and functional validation. Previously, much work in the area of CHC biosynthesis has relied on biochemical approaches and the field has largely not taken advantage of potential genetic approaches as demonstrated here. The results here provide concrete functional evidence of the roles of numerous genes in CHC biosynthesis. This paper is the product of a tremendous amount of work that will provide good leads for future biological studies. The main weakness of the manuscript is that it does not provide new mechanistic insights into any compelling biological question.

Detailed comments are as follows:

In the first section of the paper, Dembeck et al. claim to have detected 16 previously unidentified CHCs from *Drosophila melanogaster*, including dienes with double bonds in the even-numbered configurations as well as methyl-branched CHCs which are not 2-methyl-branched. The authors stated that they were able to identify these compounds because they use a thin high res column (DB-5) and a relatively long temperature program. I am concerned about the use of this column for the separation of *D. melanogaster* CHCs as this column cannot separate two major compounds in female *Drosophila*. Peak 46 contains both 2Me-C26 and 7,11-C27:2. This is a concern for the second part of the paper (the *Desat2* allele and female CHC phenotyping) because 2Me-C26 is a major compound. If they cannot separate these two compounds, then it would be impossible to quantify accurately the amount of 7,11-C27:2 in the fly. Another recent paper (Dweck et al., 2015, PNAS) also used a different method to identify CHC in *D. melanogaster*. While they identified some methyl-branched CHCs which are not 2-methyl-branched, they did not identify the dienes with the even double bonds presented in this study. The authors stated that these compounds are identified based on comparison to the reference library and to already published *D. melanogaster* CHCs and the position of the double bond was not confirmed for any of the CHCs. While most of these compounds exist in low quantities, I suggest the authors proceed with caution on claiming that they have identified new (and unusual) compounds that many other groups using different methods over the years have not been able to identify.

In the second part of the manuscript, Dembeck et al. report that the African CHC phenotype (predominantly 5,9-C27:2) is segregating in the DGRP. This section reports the mismatch between what is called the "16-bp deletion / ancestral alleles" of *Desat2* and female CHC phenotype. Two DGRP lines, DGRP_105 and DGRP_551 have the African allele ("functional *Desat2*") but the cosmopolitan phenotype. This analysis is based on protein sequence alone and does not take into account the possibility that unique polymorphisms in the *Desat2* regulatory region of these lines changed the expression of *Desat2*. Therefore, without properly quantifying whether *Desat2* is expressed or not in these lines, the authors should be careful in determining which alleles are functional based on protein sequence analysis alone (and a 16-bp deletion in the promoter that was not properly characterized previously).

In the third section which forms the bulk of the paper, the authors identify numerous genes potentially involved in CHC synthesis in *D. melanogaster*. The same authors have used similar methods to identify novel pigmentation genes in *Drosophila* (Dembeck et al., 2015, PLoS Genetics). In this section, using GWA, the authors identified genetic variants in hundreds of genes which could potentially affect CHC composition. The authors selected 24 candidate genes for functional studies. They used an oenocyte specific driver (although this driver also expresses GAL4 in some other tissues) to knockdown candidate genes in the oenocytes, where CHCs are synthesized. This is a very comprehensive section that entails a tremendous amount of work in both genomic analysis and functional characterization. The authors found that oenocyte-driven RNAi knockdown of 23 candidate genes individually showed quantitative changes in CHC profiles. These results provide very strong evidence that these candidate genes are involved in CHC biosynthesis, but there are two caveats to consider.

First, some of these genes might not be expressed in adult oenocytes. A brief look on the FlyAtlas website showed that the transcripts of some of these genes are not present or expressed at very low levels in the "adult carcass" where the oenocytes are located. Therefore, it is not known whether the phenotypes that the authors observed are off target effects of RNAi or altered physiology of the whole fly due to the oenocyte driver that they use, which may be driving GAL4 in other parts of the fly.

Second, while some of the effects of the candidate genes can be explained by the biological/molecular functions of the gene (i.e. fatty acid reductase influencing chain length), others are not (i.e. P450s affecting CHC abundance, although Qiu et al. 2012 PNAS identified the exact mechanism for another P450, *Cyp4g1* in CHC synthesis). How would a P450 (*Cyp9f2*) be involved in elongation of CHCs? As far as is known, that is not one of the reactions that a P450 could catalyze.

---

## [Author Response]

*We felt that the manuscript was very interesting but that some key factors need amendment to reflect insect biochemistry and editorial changes to increase the potential audience of this nice genetic analysis. We therefore ask you to do the following:*

*1) Rework the manuscript to properly reflect the specifics of FA biosynthesis in insects rather than rely upon the plant model of the pathway.*

We have addressed these points in the revised manuscript.

*2) Strive to present the broader picture and impacts of the results found in this study beyond CHC metabolism in* Drosophila.

We have revised the manuscript to address this comment in the Introduction and Discussion. We note that it is possible to dissect highly polygenic traits down to the functional level, in accordance with the suggestion of Reviewer 1. We also note in the Discussion that there is no need to assume that there are few causal genes for potentially adaptive traits. However, we are not the first to demonstrate that the genetic basis of high-dimensional polygenic traits can be assessed using data decomposition approaches, so we did not stress this point.

*3) Clarify that the candidate genes found may influence the CHC accumulation but further work needs to be done to test the specific enzymatic activities.*

We have added the following statement to the Discussion: “Further studies are needed to clarify the effects on CHC production of these genes and their interactions, and to test specific mechanisms and enzymatic activities through which they exert these effects.”

*4) Discuss issues on chemical identification and quantification.*

We have addressed this issue by adding a new [Supplementary-material SD10-data] which incorporates our responses to the reviewer’s comments.

*Explicit reviews can be found below.*

*Reviewer #1:*

*The authors investigate the quantitative genetic variation in CHC hydrocarbons within Drosophila using a GWA approach and identify a highly polygenic underpinning to this trait. They then validate the potential role of 24 candidate genes in altering this trait. This analysis was very nicely conducted but the writing was solely built around CHC composition in* Drosophila *which will make it difficult for this paper to attract the potential readership that it should obtain. From my reading, there are two broad messages that all researchers using GWA should be able to obtain from this manuscript.*

*1) It is possible to being dissecting highly polygenic GWA traits that have 100s or more potential candidate genes down to the functional level. There is no need to assume that there are few causal genes for potentially adaptive traits.*

*2) Polygenic adaptive traits may be structured in a manner that will allow data decomposition approaches to allow significant progress in understanding complex traits like metabolic mixtures.*

We appreciate these suggestions and have revised the manuscript to address them in the Introduction and Discussion. We explicitly note that it is possible to dissect highly polygenic traits down to the functional level and that there is no need to assume that there are few causal genes for potentially adaptive traits.

*The manuscript would likely get broader visibility if there was more general information about the bigger picture presented in the manuscript that could pull in researchers studying GWA or metabolic traits in other systems as this manuscript has information for those researchers even if they are studying plants or fungi or humans. While motivated researchers will see the broader impacts of this manuscript on the GWA/metabolite field it would help to work and bring them into the research without this effort by presenting the broader ramifications of the research in the Introduction and Discussion.*

Again, we appreciate the suggestion but feel that in this case the broader impacts with respect to other complex traits and organisms will be readily understood by a general reader, and that our CHC analysis will be of broad interest in and of itself. Since we are not the first to demonstrate that the genetic basis of high-dimensional polygenic traits can be assessed using data decomposition approaches, we did not stress this point.

*While I appreciate the use of PCs to reduce the dimensionality, this is an inherently linear approach that can have trouble with non-linear processes such as epistatic variation in biosynthetic pathways. This can be illustrated by Figure 5 where the PC1 x PC2 plot shows two linear relationships. One going NW to SE and the other from NE to SW, this suggests that PC1 and PC2 are capturing variation at potentially biosynthetic loci that work within the pathway to control structural variation. Did the authors attempt to parse out these potential biosynthetic clues to synthesize a pathway? Another reason to consider this is that the PC approach can hide key biosynthetic genes at the intersection of the two PCs. This is true in the glucosinolate work within* Arabidopsis *wherein the PC approach utilized by Bergelson and colleagues was unable to identify all of the key biosynthetic enzymes that could be identified on a more metabolite by specific metabolite analysis or alternatively using ratios of precursors to products (Wentzell et al., 2007; Chan et al., 2011; Brachi et al., 2015). While I don't feel this is necessary to conduct, it would be beneficial to discuss this difficulty that is imparted by the use of linear trait decomposition for GWA.*

*Brachi, B. et al.(2015). Coselected genes determine adaptive variation in herbivore resistance throughout the native range of Arabidopsis thaliana. Proceedings of the National Academy of Sciences of the United States of America 112, 4032-4037.*

*Chan, E.K. et al. (2011). Combining genome-wide association mapping and transcriptional networks to identify novel genes controlling glucosinolates in Arabidopsis thaliana. PLoS Biol 9, e1001125.*

*Wentzell, A.M. et al. (2007). Linking metabolic QTLs with network and cis-eQTLs controlling biosynthetic pathways. PLoS Genet 3, 1687-1701.*

We appreciate this insightful and thought-provoking comment from the reviewer. We added the following text to the Discussion regarding the motivation of using PCs and caveats of losing the ability to detect genes that control variation of individual CHCs: “We used principal component analysis to reduce dimensionality of the data, which was motivated by the high co-linearity among the CHCs. However, PCA may cause genetic variation for a certain CHC to be distributed among multiple PCs and therefore dilute its association with QTLs.”

Regarding the specific concerns raised by the reviewer:

A) PC1 and PC2 are orthogonal (uncorrelated) by definition. The appearance of certain linear relationships with subsets of lines may be a result of projecting multi(>2)-dimensional data onto the 2-D plane created by PC1 and PC2. Such patterns may or may not be biologically meaningful. However, we are not aware of a good approach that would use such patterns to synthesize a biosynthetic pathway. On the other hand, albeit incomplete and lacking hierarchical information, our approaches using individual PCs did arrive at key genes involved in the CHC biosynthetic pathways.

B) We agree with the reviewer that analysis using PCs may miss genes controlling the biosynthesis of CHCs. However, given the severe co-linearity of the data, we think the gain by reducing dimensionality is greater than the loss. Following the reviewer’s suggestion, we acknowledge this limitation in the Discussion.

*Reviewer #2:*

*This manuscript represents a great deal of very good work and can be made into an excellent publication. However, much of the biochemistry upon which it is based needs extensive correcting and rewriting. The identification of new components in this much studied species is well done and important. In the subsection “Functional validation of candidate genes”, second paragraph: Fatty acid biosynthesis begins with an acetyl-CoA and then elongation occurs with a malonyl-CoA in which the energy is supplies by decarboxylation of the malonyl-CoA as two carbons incorporated into the growing chain. The next three steps are a reduction of a carbonyl to an alcohol, dehydration, and reduction of the carbon-carbon double bond. The authors are making too extensive use of plant lipid metabolism data. The FAS in animals, including flies, occurs on the multifunctional protein fatty acid synthetase (FAS) in which the growing chain is attached to an acyl carrier type unit that is part of the multifunctional (7 enzyme activities on one protein, act in a dimer). It is not typically called an acyl carrier protein complex. Products of insect FASs, particularly* Drosophila *FAS, can be 14, 16 or 18 carbons. A thioesterase that is part of the multienzyme FAS removes the elongated chain as a free fatty acid. I do not know what is meant by conjugating and the acyl group is never removed by a thioreductase. In all cases studied in insects, desaturation and elongation take place in a microsomal fraction (endoplasmic reticulum) and not the mitochondrial fraction (many references).*

We appreciate the reviewer's expertise on this topic and re-wrote the entire paragraph as the reviewer suggested to accurately describe fatty acid biosynthesis in insects.

*In the subsection “Functional validation of candidate genes”, fourth paragraph: Fatty acids are elongated and desaturated attached to an ACP in plants. In insects, the parallel reactions occur with the acyl group attached to CoASH. The evidence to date suggests that the elongated fatty acyl-CoA is reduced to an aldehyde prior to oxidative decarbonylation, the latter step catalyzed by* Cyp4G1 *in* Drosophila *(Qiu et al., 2012). While it is possible that the elongated fatty acyl-CoA is reduced from acyl-CoA to aldehyde to alcohol and then re-oxidized to aldehyde before oxidative decarbonylation, there is no evidence for this to date in insects. (See Qiu et al. and the Reed et al. papers referenced there.)*

As the reviewer suggested, we have removed the reference to plant fatty acid elongation and changed the paragraph to accurately state what is known in insects.

*In the subsection “Functional validation of candidate genes”, end of fourth paragraph: An esterase cleaves an ester bond and does not reduce the carboxylic ester group on the fatty acyl-CoA (not ACP) to an alcohol. A FAR (fatty acyl-CoA reductase) does the reduction.*

We have removed this sentence from the manuscript.

*In the subsection “Functional validation of candidate genes”, sixth paragraph: I am not convinced the CYPs studied here are involved in CHC turnover.*

We appreciate the reviewer's skepticism. This line is our speculation based on the RNAi knockdown results. We propose it merely as a hypothesis for future functional studies of these CYPs. We have modified this statement to explicitly state that it is speculation.

In the subsection “Functional validation of candidate genes”, last paragraph: The authors are comparing 3-methyl fatty acids with a 3-methyl hydrocarbon derived from a n-3 methyl fatty acid.

We understand the reviewer’s point that for the 3-methylhydrocarbons the methyl branch is inserted right at the beginning and then elongation happens with the methyl-branched end becoming farther and farther away from the CoA “head”, whereas in a 3-methyl fatty acid the methyl group is on carbon 3 counted from the carboxyl “head”. We have removed our previous speculation from the manuscript.

*When stating “Our results provide a new perspective and highlight the complexity of the biosynthetic and catabolic pathways that regulate the dynamics of CHC composition”, the authors are blithely referring to the biosynthetic pathways of hydrocarbons (which are somewhat known) with the catabolic pathways (which to my knowledge almost nothing is known). This sentence is illustrative of several times in the manuscript where the authors make sweeping statements of biochemistry for which there is no basis.*

We have modified this sentence to say "Our results provide new perspective and highlight the complexity of the CHC dynamics."

*Reviewer #3:*

*In the first section of the paper, Dembeck et al. claim to have detected 16 previously unidentified CHCs from* Drosophila melanogaster*, including dienes with double bonds in the even-numbered configurations as well as methyl-branched CHCs which are not 2-methyl-branched. The authors stated that they were able to identify these compounds because they use a thin high res column (DB-5) and a relatively long temperature program. I am concerned about the use of this column for the separation of* D. melanogaster *CHCs as this column cannot separate two major compounds in female* Drosophila*. Peak 46 contains both 2Me-C26 and 7,11-C27:2. This is a concern for the second part of the paper (the* Desat2 *allele and female CHC phenotyping) because 2Me-C26 is a major compound. If they cannot separate these two compounds, then it would be impossible to quantify accurately the amount of 7,11-C27:2 in the fly. Another recent paper (Dweck et al., 2015, PNAS) also used a different method to identify CHC in* D. melanogaster*. While they identified some methyl-branched CHCs which are not 2-methyl-branched, they did not identify the dienes with the even double bonds presented in this study. The authors stated that these compounds are identified based on comparison to the reference library and to already published* D. melanogaster *CHCs and the position of the double bond was not confirmed for any of the CHCs. While most of these compounds exist in low quantities, I suggest the authors proceed with caution on claiming that they have identified new (and unusual) compounds that many other groups using different methods over the years have not been able to identify.*

Separation of the odd-carbon 7,11-dienes and the preceding even-carbon 2-methyl-alkane: Both separate on our column (as in Dweck et al. 2015 PNAS) by just a couple of Kovats units. So, when the diene is absent the alkane can be readily separated and vice versa. However, when the diene is a major component (e.g., 7,11-C27:2 in females, where it serves as a sex pheromone component), the alkane represents a tiny shoulder preceding the diene and they cannot be reliably separated. See our Figure 1: Peak 57 in males is 2Me-C26 and it elutes just slightly ahead of peak 57 in females, which is mainly 7,11-C27:2. Therefore, this peak can be highly variable depending whether it is in males or females. In order to treat this variable peak consistently in all chromatograms, we considered these 2 compounds together as a single peak for integration, with the recognition that it represents almost exclusively the diene in females and the alkane in males. The paper referenced by the reviewer (Dweck et al. 2015 PNAS) in fact did not separate 2Me-C26 and 7,11-C27:2 (their peaks #74 and 75 in Table Supplement 1 and Figure Supplement 1). While the table shows them separated by 3 Kovats units, the figure shows them co-eluting in females because of the large amount of #75. Only when males are examined can #74 be separated because males lack #75 (7,11-C27:2). The same is true for shorter chain compounds. In fact, Dweck et al. (2015 PNAS) confirm our findings that the 2-methyl-branched compounds are found in males (their peaks #59, 68) and either not found at all in females or in very small amounts. So, our Peak 46 contains both 2Me-C26 and 7,11-C27:2 (as in Dweck et al. 2015 PNAS peaks 74 and 75), but the alkane represents a tiny fraction of the diene that serves as sex pheromone. Dweck et al. (2015 PNAS) used a HP5 column (30 m, 0.25 mm, 0.25 µm) and ran >45 min chromatograms and could not separate these compounds in females. We ran an identical GC phase (DB-5) but in a column (20 m, 0.18 mm, 0.18 µm) that allowed high resolution in much shorter run times of 32 min to optimize the analysis of thousands (hundreds?) of chromatograms. Others (e.g., Frentiu et al. 2010 Evolution) also show no evidence of separating these compounds in females.

The reviewer is rightly concerned that in the second part of the paper (the *Desat2* allele and female CHC phenotyping) “because 2Me-C26 is a major compound”. As stated above, in normal females 2Me-C26 represents a tiny fraction of 7,11-C27:2. In the African phenotype and in females with an intact *Desat2* gene we found high 5,9-C27:2 and 5,9-C29:2 and low amounts of 7,11-dienes. The reviewer is correct that 2Me-C26 now represents a larger fraction of this combined peak. See our Figure 2, where the leading shoulder, representing 2Me-C26 (retention time 26.25 min), becomes more apparent as peak size decreases (7,11-C27:2 decreases) and 5,9-C27:2 and 5,9-C29:2 increase. Nevertheless, we contend that the potential confounding influence of 2Me-C26 is minimal because our main objective in this section was to quantify and separate the Cosmopolitan and African phenotypes, and this is readily apparent by the relationship of 7,11-C27:2 and 5,9-C27:2.

Chemical identification of methyl-branched hydrocarbons where the methyl group is near to the chain terminus was based on the relatively high abundance of indicating fragments near the molecular ion. These are produced in the EI ion source by α-cleavage at the methyl branch on the side of the near chain terminus. Hydrogen rearrangement also leads to low-weight radical ion fragments with an even weight corresponding to fragmentation on the other side of the methyl branch. Mass spectra and characteristic ions are shown in the figures in [Supplementary-material SD10-data], see A and B for 3-methyl- and 5-methylpentacosane, from a male and a female in this study, respectively. Methyl branch position on carbon 3 was also confirmed by the retention index of these hydrocarbons: they all have a retention index increment of 68–73 on the 5% phenyl-methylpolysiloxane column (peaks 17, 41, and 53 in Table 1 in the manuscript) due to the methyl branch at that particular position (Schulz, Lipids 2001, 36:637-647).

Though the position of double bond in monoalkenes was not confirmed with microderivatization reactions the retention indices were matched to that of already published monoalkenes in *D. melanogaster.* Even positions were tentatively assigned when there was a gas-chromatographic peak with a molecular ion and fragmentation pattern of a monoalkene between two peaks of monoalkenes with double bonds at consecutive odd positions, i.e. 6- was assigned in between 7- and 5- (peak 30 between 29 and 31 in Table 1).

As for alkadienes with the double bonds in even positions just a few carbons apart, we followed the same principle as described in Howard et al. (Journal of Chemical Ecology 2003, 29:961-976), namely that these compounds (as the ones with the double bond in odd positions) have characteristic fragments resulting from the fragmentation of the molecular ion where the double bonds are and from α-cleavage at the outer side of the double bonds. An example on Figure C demonstrates the above with 6,10-hexacosadiene from one of the females analyzed in this study.

We have included a new [Supplementary-material SD10-data] to summarize these points.

*In the second part of the manuscript, Dembeck et al. report that the African CHC phenotype (predominantly 5,9-C27:2) is segregating in the DGRP. This section reports the mismatch between what is called the "16-bp deletion / ancestral alleles" of* Desat2 *and female CHC phenotype. Two DGRP lines, DGRP_105 and DGRP_551 have the African allele ("functional* Desat2*") but the cosmopolitan phenotype. This analysis is based on protein sequence alone and does not take into account the possibility that unique polymorphisms in the* Desat2 *regulatory region of these lines changed the expression of* Desat2*. Therefore, without properly quantifying whether* Desat2 *is expressed or not in these lines, the authors should be careful in determining which alleles are functional based on protein sequence analysis alone (and a 16-bp deletion in the promoter that was not properly characterized previously).*

For the analysis of DGRP lines DGRP_105 and DGRP_551, we studied the DNA sequence data. While DGRP_551 does not have any obvious mutations that could disrupt *Desat2* expression, we found that DGRP_105 has many unique polymorphisms. We recently completed a gene expression analysis of 185 DGRP lines using Affymetrix tiling arrays – the manuscript reporting these results is in press at the Proceedings of the National Academy of Sciences of the USA (Huang et al., 2015). This study was done on whole young adult flies. We checked the expression of *Desat2* from this study and found that it is highly genetically variable in females (Line FDR = 1.39 E-34, *H*^2^ = 0.766) but not in males (Line FDR = 1.83E-01, not significant). We present these results below, classifying the DGRP lines as homozygous for the derived deletion (Del/Del), the ancestral insertion (Ins/Ins) and heterozygous (Ins/Del). The expression level of DGRP_105 is shown by the red data point. It is clear that the *Desat2* expression of this line overlaps that of the Del/Del genotypes, completely consistent with our CHC analysis and DNA sequence analysis showing that variants in the *Desat2* gene in this line likely result in a non-functional protein. The genotype of this line indeed matches the Cosmopolitan CHC phenotype despite the ancestral ‘functional’ insertion. Unfortunately, DGRP_551 was not among the 181 DGRP lines included in the recent gene expression analysis. We feel that further characterization of the nature of the functional variation at *Desat2* in these lines warrants many additional experiments and is outside the scope of the current manuscript.

*In the third section which forms the bulk of the paper, the authors identify numerous genes potentially involved in CHC synthesis in* D. melanogaster*. The same authors have used similar methods to identify novel pigmentation genes in* Drosophila *(Dembeck et al., 2015, PLoS Genetics). In this section, using GWA, the authors identified genetic variants in hundreds of genes which could potentially affect CHC composition. The authors selected 24 candidate genes for functional studies. They used an oenocyte specific driver (although this driver also expresses GAL4 in some other tissues) to knockdown candidate genes in the oenocytes, where CHCs are synthesized. This is a very comprehensive section that entails a tremendous amount of work in both genomic analysis and functional characterization. The authors found that oenocyte-driven RNAi knockdown of 23 candidate genes individually showed quantitative changes in CHC profiles. These results provide very strong evidence that these candidate genes are involved in CHC biosynthesis, but there are two caveats to consider.*

*First, some of these genes might not be expressed in adult oenocytes. A brief look on the FlyAtlas website showed that the transcripts of some of these genes are not present or expressed at very low levels in the "adult carcass" where the oenocytes are located. Therefore, it is not known whether the phenotypes that the authors observed are off target effects of RNAi or altered physiology of the whole fly due to the oenocyte driver that they use, which may be driving GAL4 in other parts of the fly.*

The reviewer is correct: formally, one can never disprove the potential for off-target effects of RNAi or *GAL4* driver. Only one off-target effect is reported for the RNAi constructs we tested (*Irc* has a reported off-target effect on *CG6170*). We added the following text to the Discussion to acknowledge this caveat: “In addition, we note that phenotypic changes associated with knocking down expression of a target gene with RNAi may not be causal, but a consequence of off-target effects of RNAi or the GAL4 driver. Further studies are needed to clarify the effects on CHC production of these genes and their interactions, and to test specific mechanisms and enzymatic activities through which they exert these effects.”

*Second, while some of the effects of the candidate genes can be explained by the biological/molecular functions of the gene (i.e. fatty acid reductase influencing chain length), others are not (i.e. P450s affecting CHC abundance, although Qiu et al. 2012 PNAS identified the exact mechanism for another P450,* Cyp4g1 *in CHC synthesis). How would a P450 (*Cyp9f2*) be involved in elongation of CHCs? As far as is known, that is not one of the reactions that a P450 could catalyze.*

We removed the sentence suggesting that *Cyp9f2* may be involved in the elongation of precursors of longer chain n-alkanes and monoenes in females. We later state that: “The link between these CYPs and CHC production and maintenance is unclear. However, we speculate that oxidation reactions mediated by these CYPs may regulate CHC degradation and turnover.”